# Glucose represses dendritic cell-induced T cell responses

Simon J. Lawless[1],*, Nidhi Kedia-Mehta[1],*, Jessica F. Walls[1], Ryan McGarrigle[1], Orla Convery[1], Linda V. Sinclair[2], Maria N. Navarro[3], James Murray[1] & David K. Finlay[1,4]

Glucose and glycolysis are important for the proinflammatory functions of many immune cells, and depletion of glucose in pathological microenvironments is associated with defective immune responses. Here we show a contrasting function for glucose in dendritic cells (DCs), as glucose represses the proinflammatory output of LPS-stimulated DCs and inhibits DC-induced T-cell responses. A glucose-sensitive signal transduction circuit involving the mTOR complex 1 (mTORC1), HIF1α and inducible nitric oxide synthase (iNOS) coordinates DC metabolism and function to limit DC-stimulated T-cell responses. When multiple T cells interact with a DC, they compete for nutrients, which can limit glucose availability to the DCs. In such DCs, glucose-dependent signalling is inhibited, altering DC outputs and enhancing T-cell responses. These data reveal a mechanism by which T cells regulate the DC microenvironment to control DC-induced T-cell responses and indicate that glucose is an important signal for shaping immune responses.

[1] School of Biochemistry and Immunology, Trinity Biomedical Sciences Institute, Trinity College Dublin, 152-160 Pearse Street, Dublin 2, Ireland. [2] Division of Cell Signalling and Immunology, School of Life Sciences, University of Dundee, Dow Street, Dundee DD1 5EH, Scotland, UK. [3] Departamento Medicina/Universidad Autónoma de Madrid, Instituto Investigación Sanitaria/Hospital Universitario de la Princesa, C/Diego de Léon, 62, Madrid 28006, Spain. [4] School of Pharmacy and Pharmaceutical Sciences, Trinity Biomedical Sciences Institute, Trinity College Dublin, 152-160 Pearse Street, Dublin 2, Ireland. * These authors contributed equally to this work. Correspondence and requests for materials should be addressed to D.K.F. (email: finlayd@tcd.ie).

Cellular metabolism has emerged as an important regulator of immune cell function not only through facilitating the energy and biosynthetic demands of the cell but also through directly controlling important immune cell functions[1]. Glycolysis is important for the proinflammatory functions of multiple immune cell subsets; glycolytic enzymes and metabolites can regulate immune signalling and effector functions[2,3]. The mammalian target of rapamycin complex 1 (mTORC1) is described in many immune cells as a central regulator of immune cell metabolism that promotes elevated levels of glycolysis through promoting the activity of the transcription factors cMyc and hypoxia-inducible factor 1α (HIF1α), which induce the expression of glucose transporters and glycolytic enzymes[4–7]. mTORC1 is important in controlling the differentiation and function of immune cells, and it is becoming clear that this is achieved in part through the regulation of cellular metabolic pathways[4,8,9]. Although the mTORC1-specific inhibitor rapamycin was originally characterized as a potent immunosuppressant required for the generation of effector T-cell responses, inhibition of mTORC1 in myeloid cells actually results in increased inflammatory outputs[10,11]. Therefore, mTORC1 signalling can be either proinflammatory or anti-inflammatory depending on the immune cell subset, although it is not clear whether mTORC1-controlled metabolic alterations are important for these differential effects.

Dendritic cells (DCs) undergo substantial changes in function following immune activation to adopt an important role in stimulating immune responses, and these functional changes are associated with altered metabolism. In DCs differentiated from bone marrow in the presence of the growth factor granulocyte macrophage colony-stimulating factor (GM-CSF) (GM-DCs), rates of cellular glycolysis are rapidly increased, within minutes, once activated with lipopolysaccharide (LPS). Then over the course of 18 h, GM-DCs switch to a highly glycolytic metabolism; GM-DCs display increased glycolytic rates and an inactivation of mitochondrial oxidative phosphorylation (OXPHOS)[12,13]. At this point postactivation (18 h), DCs would normally have reached the draining lymph node where they would be interacting with T cells. The balance between glycolysis and OXPHOS is an important effector of immune cell differentiation and the modulation of inflammatory responses[14–16]. Although there is some evidence that OXPHOS levels affect DC function, the relationship between DC metabolism and DC-induced T-cell responses is not well defined[17].

As the flux through cellular metabolic pathways is controlled by the supply of nutrients, there is renewed interest in nutrient levels in immune microenvironments and how they affect immune responses. For instance, reduced glucose levels in the tumour microenvironment can directly impact upon T-cell receptor signalling and inhibit antitumour T-cell responses[3]. It is likely that nutrient levels will also be important for immune cell function at sites of bacterial and viral infections where there is considerably increased demand for nutrients, such as glucose[18,19]. DCs experience diverse microenvironments within tissue, at inflammatory sites and as they migrate to the draining lymph nodes where they activate T cells, often interacting with numerous T cells at a time[20,21]. It is not clear how the availability of nutrients within these microenvironments affects DC metabolic pathways to control DC function and the induction of T cells' responses.

Here we establish that glucose represses DC inflammatory outputs and thus DC-induced T-cell proliferation and interferon-γ (IFNγ) production. A complex glucose-sensing mTORC1/HIF1α/inducible nitric oxide synthase (iNOS) signalling circuit integrates information about glucose levels in the local microenvironment to coordinate DC metabolism and function. Competitive uptake of glucose by activated T cells can starve DCs of glucose, inactivate this glucose-sensing signalling circuit and promote proinflammatory DC outputs to enhance T-cell responses.

## Results

**Glucose deprivation enhances DC-induced T-cell responses.** Increased glycolysis is required immediately following LPS activation of DCs to facilitate an expansion of the biosynthetic machinery, that is, the Golgi and endoplasmic reticulum apparatus[22]. However, elevated glycolysis is also a feature of DC metabolism for prolonged periods after activation at points when DCs interact with T cells and promote T-cell responses. To establish the role that glucose and increased rates of glycolysis play for DC-induced T-cell responses, GM-DCs were switched from glucose into galactose 8 h after LPS stimulation. Galactose is an alternative cellular fuel to glucose that can be metabolized by glycolysis and OXPHOS to provide energy for the cell but galactose can only maintain low rates of glycolysis in GM-DCs (Fig. 1a)[2,23]. Removing glucose and limiting the rate of glycolysis in this way affected the expression of the costimulatory molecules CD80 and CD86 on activated GM-DCs. LPS-induced expression of CD80 and CD86 peaked at 24 h and then declined 48 and 72 h after stimulation when GM-DCs are cultured with glucose (Fig. 1b,c). In contrast, glucose deprived GM-DCs, that is, cells cultured in galactose, maintained high levels of CD80 and CD86 for the duration of the 72 h post LPS stimulation (Fig. 1b,c). During this time course, there were no differences in GM-DC viability (Fig. 1d). Levels of another costimulatory molecule CD40 were unchanged in the absence of glucose (Fig. 1e). Cytokines produced by DCs are another important signal for shaping the T-cell immune response. Glucose-deprived GM-DCs expressed elevated levels of *Il12a* mRNA but normal levels of *Il10* mRNA (Fig. 1f). Similar increases in *Il12a* mRNA were observed when GM-DC were cultured in decreasing concentrations of glucose (Fig. 1g). Next, the impact of these functional changes on DC-stimulated T-cell responses was investigated. GM-DCs were LPS-stimulated, pulsed with the peptide SIINFEKL and cultured in the presence of glucose or galactose for up to 72 h. At the indicated time points after LPS activation, GM-DCs from each condition were washed, placed into glucose-containing media and then co-cultured with purified CD8[+] OTI T cells for a further 48 h. CD8[+] OTI transgenic T cells express a single T-cell receptor that recognizes major histocompatibility complex I-bound SIINFEKL. GM-DCs cultured in glucose or galactose activated T cells equivalently as determined by T-cell blastogenesis and increased CD69 expression (Fig. 2a,b). However, distinct differences in the nature of the T-cell response were observed in the absence of glucose. The capacity of glucose-cultured GM-DCs to induce OTI T-cell clonal expansion declined at 48 h and was lost at 72 h post LPS stimulation (Fig. 2c,d). In contrast, GM-DCs cultured in galactose retained the ability to induce T-cell clonal expansion for the duration of the 72 h stimulation (Fig. 2c,d). Glucose-deprived GM-DCs also induced increased IFNγ production in T cells 72 h post LPS stimulation (Fig. 2e–g). A direct effect of galactose on the phenotype of LPS-stimulated GM-DC was excluded as GM-DCs cultured in glucose or glucose plus galactose were phenotypically identical and stimulated equivalent T-cell responses (Supplementary Fig. 1). Overall, the data show that glucose negatively impacts upon GM-DC-induced T-cell responses.

**Glucose controls metabolic signal transduction pathways.** The data showed that between 8 and 24 h post LPS-stimulation

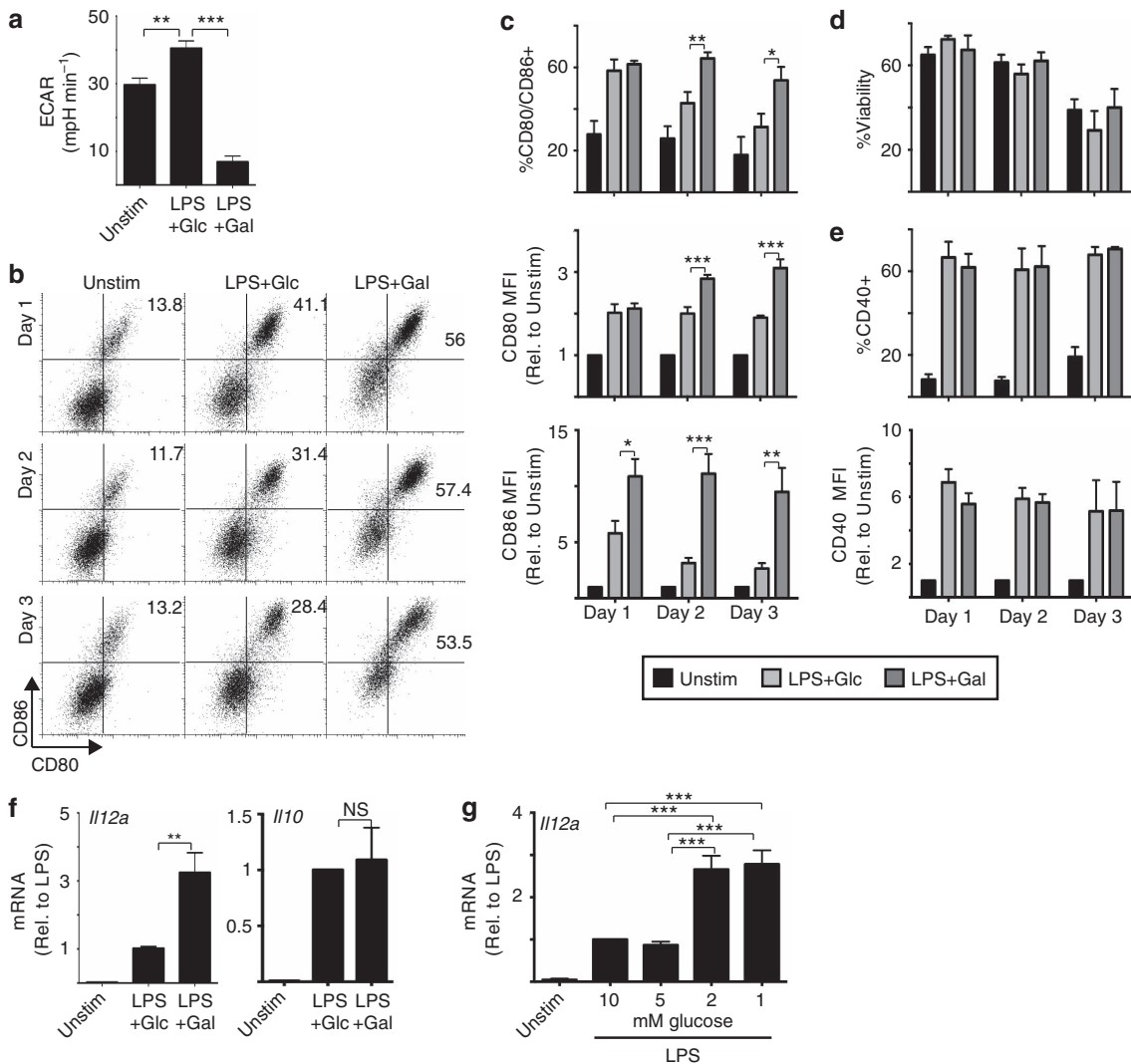

**Figure 1 | Glucose represses GM-DCs expression of costimulatory molecules and IL-12.** GM-DCs were left unstimulated (Unstim) or stimulated with LPS for 8 h, washed and then cultured in media containing either 10 mM glucose (Glc) or 10 mM galactose (Gal) (**a–f**) or in different concentrations of glucose (**g**). GM-DCs were then maintained in these culture conditions for up to 3 days as indicated. GM-DCs were analysed for (**a**) rates of glycolysis (ECAR) by seahorse analysis on day 1; by flow cytometry for (**b,c**) the expression of CD80 and CD86 costimulatory molecule expression, (**d**) cell viability by FSC/SSC analysis, (**e**) the expression of CD40 or (**f,g**) by qPCR on day 1 for *IL12a* and *IL10* mRNA expression, as indicated. Data are mean ± s.e.m. or representative of three (**a,f**), five (**g**) or six (**b–d**) separate experiments. qPCR performed in triplicate; seahorse analysis performed in quadruplicate. Data were analysed using a one-way analysis of variance with Tukey's post test (\*$P < 0.05$, \*\*$P < 0.01$, \*\*\*$P < 0.001$). MFI, mean fluorescent intensity.

GM-DCs undergo glycolytic reprogramming, the increased expression of glucose transporters and glycolytic enzymes (Fig. 3a). A key observation was that in addition to directly limiting the rate of glycolysis, switching GM-DCs from glucose to galactose also prevented glycolytic reprogramming of LPS-stimulated GM-DCs (Fig. 3b). These data argue that glucose affects metabolic signal transduction pathways. The transcription factors cMyc and HIF1α are both described to promote the expression of glucose transporters and glycolytic enzymes in immune cells[5,6,24]. As cMyc is not expressed in DCs[25], we investigated whether glucose is required for the activity of HIF1α in LPS-stimulated GM-DCs. LPS stimulation of GM-DCs resulted in the increased expression of HIF1α protein but not that of *Hif1a* mRNA (Fig. 3c, Supplementary Fig. 2a). Increased HIF1α protein corresponded to elevated HIF1α transcriptional activity, based on the expression of prolyl hydroxylase 3 (*Phd3*) mRNA, an established HIF1α target gene (Fig. 3c). The specificity of these assays was confirmed using HIF1α-null GM-DC (*Hif1a*$^{-/-}$) from *Hif1a*$^{flox/flox}$ *Vav-Cre* mice; these GM-DCs do not express

HIF1α protein or *Phd3* mRNA (Fig. 3c). LPS-induced HIF1α activity was observed 16 h after stimulation and so correlated with LPS-induced glycolytic reprogramming (Fig. 3a, Supplementary Fig. 2b). To determine whether HIF1α expression or activity was sensitive to glucose availability, LPS-stimulated GM-DCs were cultured in decreasing concentrations of glucose and HIF1α protein levels and *Phd3* mRNA levels ascertained. Reducing the concentration of glucose from 10 to 2 mM was sufficient to prevent LPS-induced HIF1α protein expression and activity (Fig. 3d,e). A second complementary approach that involved replacing glucose with galactose also prevented LPS-stimulated HIF1α expression and activity (Fig. 3e,f). However, interestingly, these two approaches regulate HIF1α protein expression through distinct mechanisms. Decreased glucose concentrations activated the AMP-activated protein kinase (AMPK), as measured by increased phosphorylation of the AMPK substrate acetyl-CoA carboxylase (ACC; Fig. 3e). AMPK is a kinase that is activated when cells are experiencing energy stress. In CD8 T cells, glucose deprivation activates AMPK to result in the inhibition of

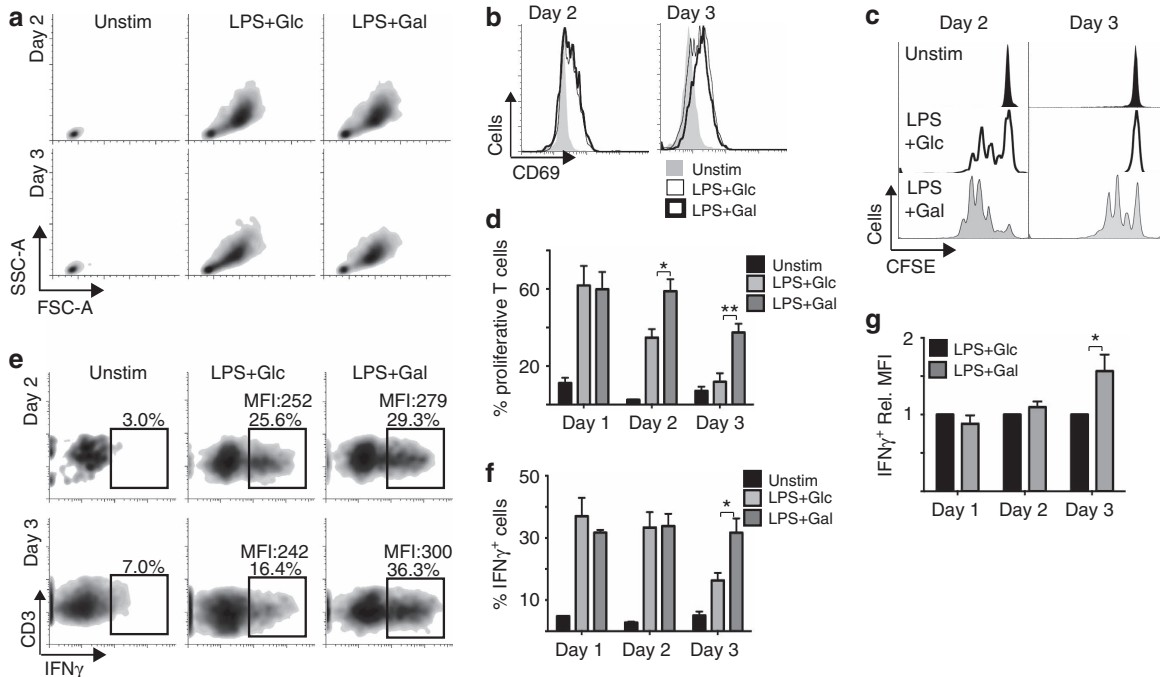

**Figure 2 | Glucose represses GM-DC-induced CD8 T-cell proliferation and IFNγ production.** GM-DCs were pulsed with SIINFEKL peptide for 8 h +/− LPS (100 ng ml$^{-1}$), washed and placed in media containing either 10 mM glucose (Glc) or 10 mM galactose (Gal). GM-DCs were then maintained in these culture conditions for 1, 2 or 3 days. On the indicated day post LPS stimulation, GM-DCs were washed and normal media containing 10 mM glucose was added before the addition of purified carboxyfluorescein succinimidyl ester (CFSE)-labelled CD8 OT-I T cells. After a co-culture period of 48 h, the OT-I T cells were analysed by flow cytometry for (**a**) cell size, (**b**) CD69 expression, (**c,d**) proliferation as measured by CFSE dilution and (**e,f,g**) IFNγ production following phorbol 12-myristate 13-actetate and ionomycin treatment for 4 h; (**e,f**) %IFNγ-positive T cells and (**e,g**) the MFI of IFNγ production in IFNγ+ T cells. Data are mean ± s.e.m. or representative of six separate experiments. Data were analysed using a one-way analysis of variance with Tukey's post-test (*$P<0.05$, **$P<0.01$). MFI, mean fluorescent intensity, Unstim, unstimulated.

mTORC1 signalling[26]. Indeed, mTORC1 is also inactive in GM-DCs cultured in low glucose, as measured by the phosphorylation of the mTORC1 substrate p70 S6-kinase (pS6K) and the S6K substrate S6 ribosomal protein (pS6) (Fig. 3e). As our previous work in CD8 T cells showed that mTORC1 activity is essential for HIF1α expression, it was reasoned that mTORC1 inactivation in these GM-DCs leads to loss of HIF1α activity[5]. Indeed, as predicted the mTORC1 inhibitor rapamycin abolishes HIF1α protein expression and activity in LPS-stimulated GM-DCs (Fig. 3g). Therefore, decreasing glucose levels inhibits HIF1α activity via the inactivation of mTORC1 signalling. In contrast, replacing glucose with galactose does not activate AMPK nor does it inactivate mTORC1 signalling (Fig. 3e). This is because, while culturing GM-DCs in galactose only maintains low levels of glycolysis, it results in increased rather than decreased rates of OXPHOS coupled to ATP synthesis, which is sufficient to avoid energy crisis and AMPK activation (Fig. 3h). Therefore, an alternative mTORC1-independent mechanism must underpin the lack of HIF1α protein expression under these conditions.

$Hif1a^{-/-}$ GM-DCs were then analysed to establish whether HIF1α is required for LPS-induced glycolytic reprogramming. LPS-stimulated $Hif1a^{-/-}$ GM-DCs had reduced rates of glycolysis and failed to upregulate the expression of glycolytic genes (Fig. 4a,b). Similarly, inhibition of mTORC1, which blocks HIF1α expression, was shown to prevent elevated glycolysis and glycolytic reprogramming in LPS-activated GM-DC (Fig. 4c,d). Therefore, the data show that glucose is required for LPS-induced glycolytic reprogramming because it facilitates the expression and activity of the HIF1α transcription factor.

**HIF1α and iNOS signalling coordinates DC metabolic pathways.** Our metabolic analysis of LPS-stimulated GM-DCs revealed that glucose is required for the inactivation of OXPHOS; galactose-cultured GM-DCs failed to downregulate OXPHOS (Fig. 5a). Previous reports have shown that nitric oxide (NO) produced by iNOS is responsible for the inactivation of OXPHOS in LPS-stimulated GM-DCs[13]. Therefore, we considered whether glucose is required for iNOS expression and NO production in LPS-activated GM-DCs. The data showed that LPS-induced NO production was completely blocked in galactose-cultured GM-DCs, as determined by measuring nitrite levels in the culture medium (Fig. 5b). Given that galactose-cultured GM-DCs have lost HIF1α expression, we investigated whether the loss of iNOS expression may be related to defective HIF1α activity. iNOS expression and activity was measured under multiple experimental conditions that inhibit HIF1α expression. Rapamycin-treated GM-DCs that do not express HIF1α (Fig. 3g) are also deficient for iNOS expression, NO production and they do not inactivate OXPHOS (Fig. 5c,d). $Hif1a^{-/-}$ GM-DC have greatly reduced levels of $Nos2$ mRNA and protein and produce reduced levels of NO (Fig. 5e,f). Therefore, the data argue that HIF1α activity is required for iNOS expression in LPS-stimulated GM-DCs. We next considered whether HIF1α expression is sufficient to promote iNOS expression. A family of prolyl hydroxylases target HIF1α for ubiquitin-mediated proteasomal degradation, thus repressing HIF1α protein levels. HIF1α protein levels increase when these prolyl hydroxylases are inhibited. While GM-DCs lacking mTORC1 activity do not express HIF1α or iNOS (Fig. 5c), promoting the stabilization of HIF1α protein with the prolyl hydroxylase inhibitor dimethyloxaloylglycine (DMOG) is sufficient to induce $Nos2$ mRNA and protein

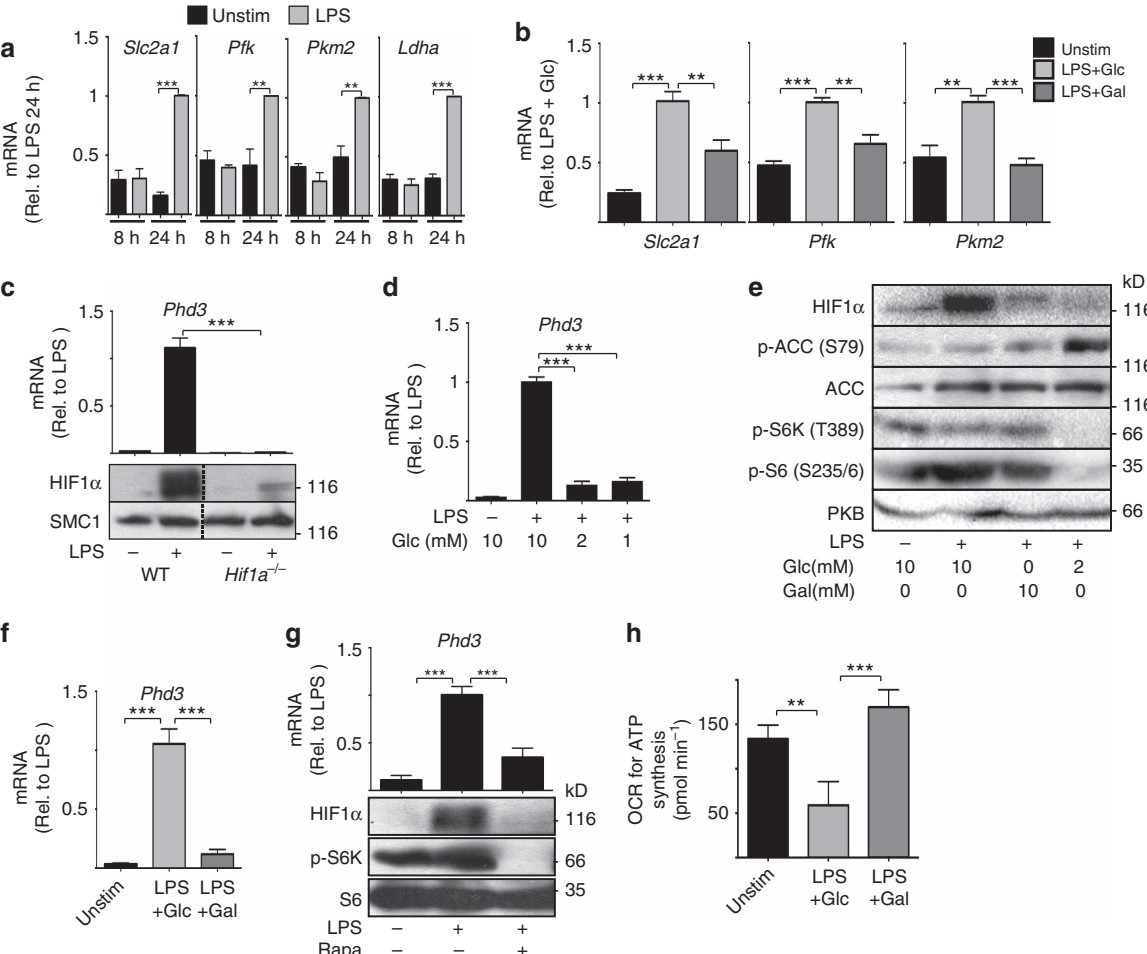

**Figure 3 | Glucose signals via mTORC1 and HIF1α to promote metabolic reprogramming of LPS-activated GM-DCs. (a)** GM-DCs were left unstimulated (Unstim) or stimulated with LPS (100 ng ml$^{-1}$) for 8 h or 24 h and analysed by qPCR for the expression of the glucose transporter *Slc2a1* and key glycolytic enzymes: phosphofructose kinase (*Pkf*), pyruvate kinase 2 (*Pkm2*), and lactate dehydrogenase a (*Ldha*). **(b)** GM-DCs were treated +/− LPS (100 ng ml$^{-1}$) for 8 h, washed and then cultured in media containing 10 mM glucose (Glc) or 10 mM galactose (Gal) for 20 h and *Slc2a1*, *Pkf* and *Pkm2* mRNA expression was measured. **(c)** GM-DCs generated from *Hif1a*$^{flox/flox}$ (WT) or *Hif1a*$^{flox/flox}$ VavCre (*Hif1a*$^{-/-}$) mice were treated +/− LPS (100 ng ml$^{-1}$) for 24 h, and then HIF1α protein and *Phd3* mRNA levels were measured. **(d)** qPCR analysis of *Phd3* mRNA levels in GM-DCs treated +/− LPS (100 ng ml$^{-1}$) for 8 h in normal media and then for 20 h in media with different glucose concentrations. **(e)** GM-DCs were treated +/− LPS (100 ng ml$^{-1}$) for 8 h, washed and then cultured in media containing different concentrations of glucose or galactose for 20 h before immunoblot analysis for HIF1α, phosphorylated and total acetyl-CoA carboxylase (pACC and ACC), phosphorylated p70 S6-kinase (p-S6K), phosphorylated S6 ribosomal protein (p-S6) and total protein kinase B (PKB, loading control). **(f)** GM-DCs were treated +/− LPS (100 ng ml$^{-1}$) for 8 h, washed and then cultured in media containing either 10 mM glucose (Glc) or 10 mM galactose (Gal) for 20 h and analysed by qPCR for the expression of *Phd3* mRNA. **(g)** GM-DCs were treated +/− LPS (100 ng ml$^{-1}$) +/− rapamycin (20 nM) for 20 h and analysed by immunoblot analysis for HIF1α, p-S6K and total S6 and by qPCR for *Phd3* mRNA. **(h)** GM-DCs were treated +/− LPS (100 ng ml$^{-1}$) for 8 h, washed and then cultured in media containing either 10 mM glucose or 10 mM galactose for 20 h and analysed for rates of OXPHOS coupled to ATP synthesis. Data are mean ± s.e.m. of at least three separate experiments. Representative immunoblot of at least three separate experiments are shown. Data were analysed using a one-way analysis of variance with Tukey's post test (**$P < 0.01$, ***$P < 0.001$). OCR, oxygen consumption rate.

expression and to increase NO production (Fig. 5g,h). Interestingly, experiments that disrupted iNOS activity in LPS-stimulated GM-DCs revealed that HIF1α protein expression is also dependent on iNOS activity. Direct inhibition of iNOS, using the specific inhibitor S-ethylisothiourea (SEITU), prevented HIF1α protein expression and activity in LPS-stimulated GM-DCs (Fig. 6a). NO production by iNOS requires the substrate arginine. Depriving GM-DCs of arginine prevented NO production in LPS-stimulated GM-DC and inhibited HIF1α activity (Fig. 6b,c). Arginine deprivation did not inhibit mTORC1 signalling arguing that reduced HIF1α activity was due to decreased NO production (Fig. 6c). Consistent with decreased HIF1α activity, arginine deprivation prevented LPS-induced glycolysis and glycolytic reprogramming (Fig. 6d,e). Finally,

GM-DCs were generated from iNOS knockout mice; these cells did not express HIF1α protein or *Phd3* mRNA in response to LPS stimulation (Fig. 6f). Reactive oxygen species and reactive nitrogen species are known inhibitors of the prolyl hydroxylases that target HIF1α for degradation. Therefore, the data suggest that NO is required for HIF1α stabilization. The data show that HIF1α and iNOS have a reciprocal relationship, and each molecule is required for the expression of the other. A time course analysis of *Nos2* mRNA expression and HIF1α activity (*Phd3* mRNA expression) revealed that *Nos2* mRNA expression but not HIF1α activity is increased 4 h after LPS activation. There is an additional increase in *Nos2* mRNA expression observed after 16 h that coincides with the induction of *Phd3* expression, that is, HIF1α activity

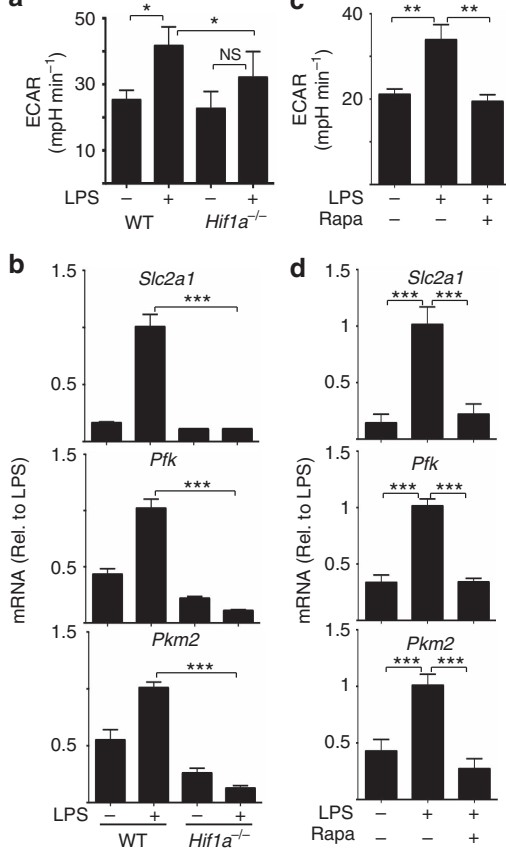

**Figure 4 | mTORC1 and HIF1α signalling is required for metabolic reprogramming in activated GM-DCs.** (**a,b**) Hif1a[flox/flox] (WT) or Hif1a[flox/flox] VavCre (Hif1a[−/−]) GM-DCs were left unstimulated or stimulated with LPS (100 ng ml[−1]) for 20 h and analysed for rates of glycolysis (**a**) or by qPCR for the mRNA expression of the glucose transporter Slc2a1 and glycolytic enzymes phosphofructose kinase (Pkf) and Pyruvate kinase 2 (Pkm2) (**b**). (**c,d**) GM-DCs were left unstimulated or stimulated with LPS (100 ng ml[−1]) +/− rapamycin (Rapa, 20 nM) for 20 h and analysed for rates of glycolysis (**c**) or by qPCR for the mRNA expression Slc2a1 and Pkf and Pkm2 (**d**). Data are mean ± s.e.m. at least three separate experiments performed in quadruplicate (**a,c**) or triplicate (**b,d**). Data were analysed using a one-way analysis of variance with Tukey's post test (*P < 0.05, **P < 0.01, ***P < 0.001). ECAR, extracellular acidification rate.

(Supplementary Fig. 2b,c). These data support a model where iNOS-dependent NO initially promotes HIF1α protein stabilization and then a feed-forward loop ensues where HIF1α promotes iNOS expression and NO production stabilizes HIF1α protein, leading to elevated levels of both iNOS and HIF1α. Nos2 mRNA expression only becomes sensitive to rapamycin once HIF1α is active, after 16 h, arguing that mTORC1 controls the HIF1α-iNOS signalling axis by regulating the expression of HIF1α (Supplementary Fig. 2b,c).

Given that classical DC subsets do not express iNOS, we next considered whether NO from exogenous sources, such as from activated macrophages, would be sufficient to stabilize HIF1α within activated GM-DCs that lack iNOS expression. The addition of a NO donor to LPS-stimulated Nos2[−/−] GM-DCs for 4 h was sufficient to induce the expression of HIF1α protein and activity (Fig. 6f). Co-culture experiments were also performed using a transwell system where molecules such as NO, but not cells, can be exchanged between the chambers. The addition of

transwells containing LPS/IFNγ-activated bone marrow-derived macrophages (BMDMs) to wells containing LPS-activated Nos2[−/−] GM-DCs for 4 h was sufficient to induce HIF1α protein expression and activity in the Nos2[−/−] GM-DCs (Fig. 6g). This effect was dependent on BMDMs expressed iNOS as the HIF1α protein expression in Nos2[−/−] GM-DC was lost when the iNOS inhibitor SEITU was added to the coculture (Fig. 6h). These data demonstrate that a complex signalling circuit involving mTORC1, HIF1α, iNOS and NO coordinates the LPS-induced metabolic shift from OXPHOS to glycolysis.

**HIF1α negatively affects DC-induced T-cell responses.** To prove that the anti-inflammatory effect of glucose is mediated by HIF1α/iNOS signalling, the functions of Hif1a[−/−] and Nos2[−/−] GM-DCs were investigated. LPS-stimulated Hif1a[−/−] and Nos2[−/−] GM-DCs sustained elevated levels of costimulatory molecules CD80 and CD86 as seen in galactose-cultured GM-DCs (Fig. 7a–d, Supplementary Fig. 3c). The addition of exogenous NO to Nos2[−/−] GM-DC cultures, which is sufficient to induce HIF1α protein expression (Fig. 6f), prevented these elevated levels of CD80 and CD86 expression (Fig. 7c,d, Supplementary Fig. 3b,c). Hif1a[−/−] and Nos2[−/−] GM-DCs also had increased expression of IL12a and TNF mRNA but normal IL10 mRNA expression (Fig. 7e–g). Similar increases in IL12a mRNA levels were observed under other experimental conditions that resulted in reduced HIF1α expression: GM-DCs treated with rapamycin or SEITU (Supplementary Fig. 3d,e) or GM-DCs deprived of glucose (Figs 1f and 3f). Exposing LPS-stimulated Nos2[−/−] GM-DCs to exogenous NO for just 4 h was sufficient to inhibit the expression of IL12a and TNF mRNA (Fig. 7f,g). Additionally, pharmacologically increasing HIF1α protein levels in Nos2[−/−] GM-DCs using DMOG was sufficient to inhibit IL12a mRNA expression, thus confirming that HIF1α negatively regulates the production of this proinflammatory cytokine (Fig. 7h).

Consistent with these observed differences in GM-DCs proinflammatory functions, Hif1a[−/−] and Nos2[−/−] GM-DCs had enhanced capacity to induce OTI T-cell proliferation (Fig. 7i–l). Exogenous NO reversed the enhanced T-cell proliferation induced by Nos2[−/−] GM-DCs (Fig. 7k,l). Together, these data argue that glucose-controlled HIF1α represses GM-DCs proinflammatory functions and limits DC-dependent T-cell responses.

**T cells can deplete glucose from the DC microenvironment.** This study has characterized a complex signalling circuit in GM-DCs that is sensitive to the available levels of glucose but also to the availability of other nutrients (Fig. 8a). Amino acid availability can also impact upon this signalling circuit through inhibiting NO production or mTORC1 activity (Fig. 6b, Supplementary Fig. 4a). Systemic levels of glucose are tightly controlled and glucose levels do not drop to low levels even in the starved state; this is not surprising given the importance of glucose as a cellular fuel. However, there is a growing appreciation that glucose levels can become limiting in discrete microenvironments, such as the tumour or inflammatory microenvironments. Close interactions with T cells are central to the function of DCs to stimulate T-cell responses. Given that during immune activation T cells have a substantially increased demand for glucose and the fact that multiple activating T cells can form clusters around a single DC[20,21], we reasoned that these circumstances could result in glucose becoming limiting within the immediate DC microenvironment. To investigate whether activating T cells could limit the nutrient levels available to DCs, we measured the uptake of the fluorescent glucose analogue 2-(N-(7-nitrobenz-2-oxa-1,3-diazol-4-yl)amino)-2-deoxyglucose

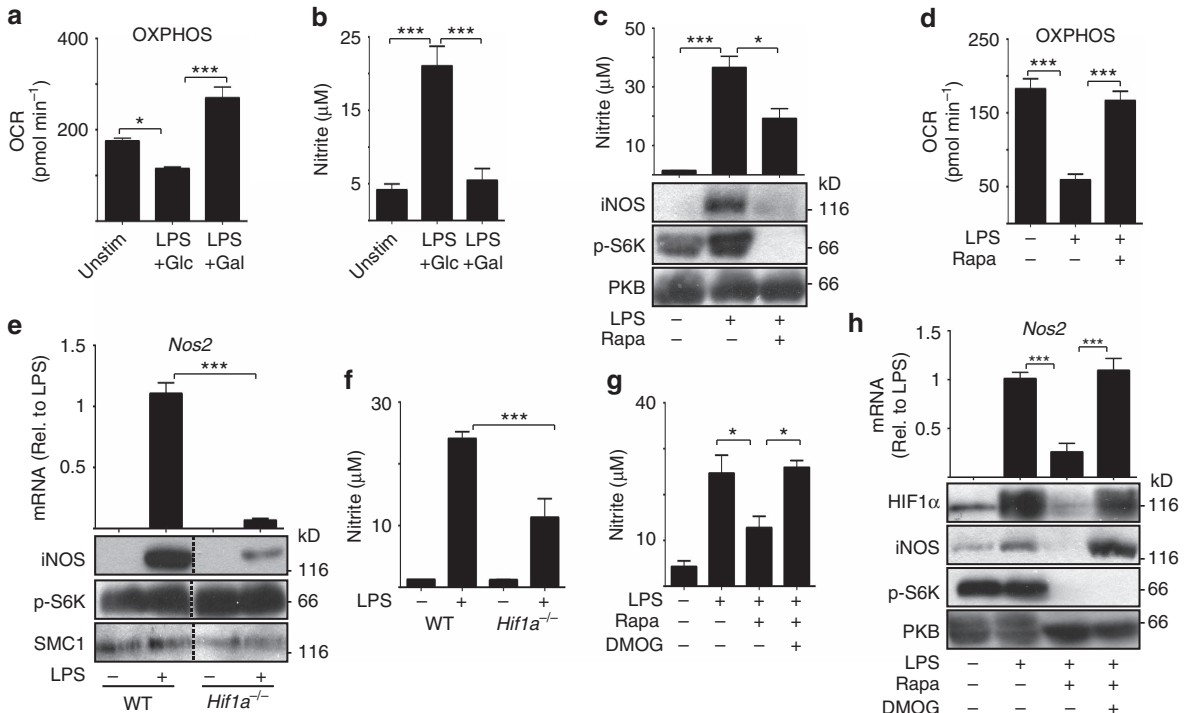

**Figure 5 | mTORC1 and HIF-1α signalling is required for iNOS activity in LPS-activated GM-DCs.** GM-DCs were left unstimulated (Unstim) or stimulated with LPS (100 ng ml$^{-1}$) for 8 h, washed and then cultured in media containing either 10 mM glucose (Glu) or 10 mM galactose (Gal) for 20 h prior to the measurement of (**a**) OXPHOS levels by seahorse analysis and (**b**) nitrite production by the Greiss reaction. (**c,d**) GM-DCs were left unstimulated or stimulated with LPS (100 ng ml$^{-1}$) +/− rapamycin (20 nM) for 20 h and analysed for (**c**) nitrite production by the Greiss reaction (upper panel) and by immunoblot analysis (lower panel) for protein levels (phosphorylated p70 S6-kinase, p-S6K; protein kinase B, PKB) or (**d**) OXPHOS levels. (**e,f**) Hif1a$^{flox/flox}$ (WT) or Hif1a$^{flox/flox}$ VavCre (Hif1a$^{-/-}$) GM-DCs were left unstimulated or stimulated with LPS (100 ng ml$^{-1}$) for 20 h, then analysed (**e**) by qPCR for Nos2 mRNA expression (upper panel) and by immunoblot analysis (lower panel) for protein levels (Structural Maintenance Of Chromosomes protein, SMC1) or (**f**) for nitrite production by the Greiss reaction. (**g,h**) GM-DCs were left unstimulated or stimulated with LPS (100 ng ml$^{-1}$) +/− rapamycin (20 nM) +/− DMOG (200 μM) for 20 h and analysed for (**g**) nitrite production by the Greiss reaction or (**h**) by qPCR for Nos2 mRNA expression (upper panel) and immunoblot analysis for protein levels (lower panel). Data are mean ± s.e.m. of at least three separate experiments performed in quadruplicate (**a,d**) or triplicate (**b,c,e–h**). Representative immunoblots of at least two separate experiments are shown. Data were analysed using a one-way analysis of variance with Tukey's post test (*P < 0.05, ***P < 0.001). OCR, oxygen consumption rate.

(NBDG) into T cells and GM-DCs in co-culture experiments. SIINFEKL-pulsed GM-DCs were co-cultured with increasing numbers of purified CD8 OTI T cells for 20 h and NBDG uptake analysed by flow cytometry. Increasing the ratio of T cells to DCs resulted in decreased levels of NBDG uptake into GM-DCs (Fig. 8b). In contrast, the level of NBDG uptake into the T cells was equivalent for all DC:T-cell ratios, indicating that there was not a global deficit in NBDG in these cultures (Fig. 8b, Supplementary Fig. 4b). These data argue that T cells can limit glucose uptake into GM-DCs due to competitive uptake. Given that glucose deprivation can result in the inhibition of mTORC1 signalling (Fig. 3e)[26], we considered whether mTORC1 signalling was altered in GM-DCs in these co-culture experiments. Increasing numbers of T cells resulted in decreased levels of pS6 within the GM-DCs but not in T cells (Fig. 8c, Supplementary Fig. 4c). pS6 levels were also investigated in GM-DCs by confocal microscopy. LPS-stimulated GM-DCs cultured at a 1:2 DC:T-cell ratio were observed to have high levels of pS6 (white arrows), as compared to the rapamycin-negative control (Fig. 8d,e). In contrast, when cultured at a 1:10 DC:T-cell ratio, T cells were observed to cluster around GM-DCs and pS6 levels within the GM-DCs were comparable to the rapamycin control (yellow arrows; Fig. 8d,e). The T cells within these DC:T-cell clusters maintained high levels of pS6, indicating that the T cells were nutrient replete (Fig. 8d,e). Within the 1:10 DC:T-cell ratio cultures, there were a minority of GM-DCs that

were only interacting with a small number of T cells and these GM-DCs had high levels of pS6 (white arrow compared to yellow arrow; Fig. 8f). Therefore, the data suggest that local nutrient depletion in the GM-DC microenvironment results in the loss of mTORC1 signalling as opposed to a global deficit of nutrients in the cultures (Fig. 8d–f). According to the signalling circuit identified in this study (Fig. 8a), reduced mTORC1 signalling would be predicted to result in decreased HIF1α and iNOS activity. As CD8 T cells do not produce NO, we determined the level of NO production in these cultures as a measure of HIF1α/iNOS signalling in the GM-DCs. There was a clear decrease in NO production in the co-cultures with increased CD8 T cells (Fig. 8g). Consistent with a role for glucose and mTORC1/HIF1α/iNOS signalling in repressing DC-induced T-cell responses, there was increased IFNγ production in OTI T cells activated under conditions where nutrient availability to GM-DCs was limited (Fig. 8h, Supplementary Fig. 4d,e). Numerous studies have demonstrated that pharmacological inhibition of mTORC1, using the inhibitor rapamycin, increased the proinflammatory outputs of human and murine DC subsets that do not express iNOS[27–30]. This suggested to us that the inhibition of mTORC1 due to nutrient deprivation in iNOS-negative DC would also lead to enhanced T-cell responses. To test this hypothesis, Nos2$^{-/-}$ GM-DCs were co-cultured with different ratios of OTI T cells, as above, and IFNγ expression in the T cells was assessed. As predicted,

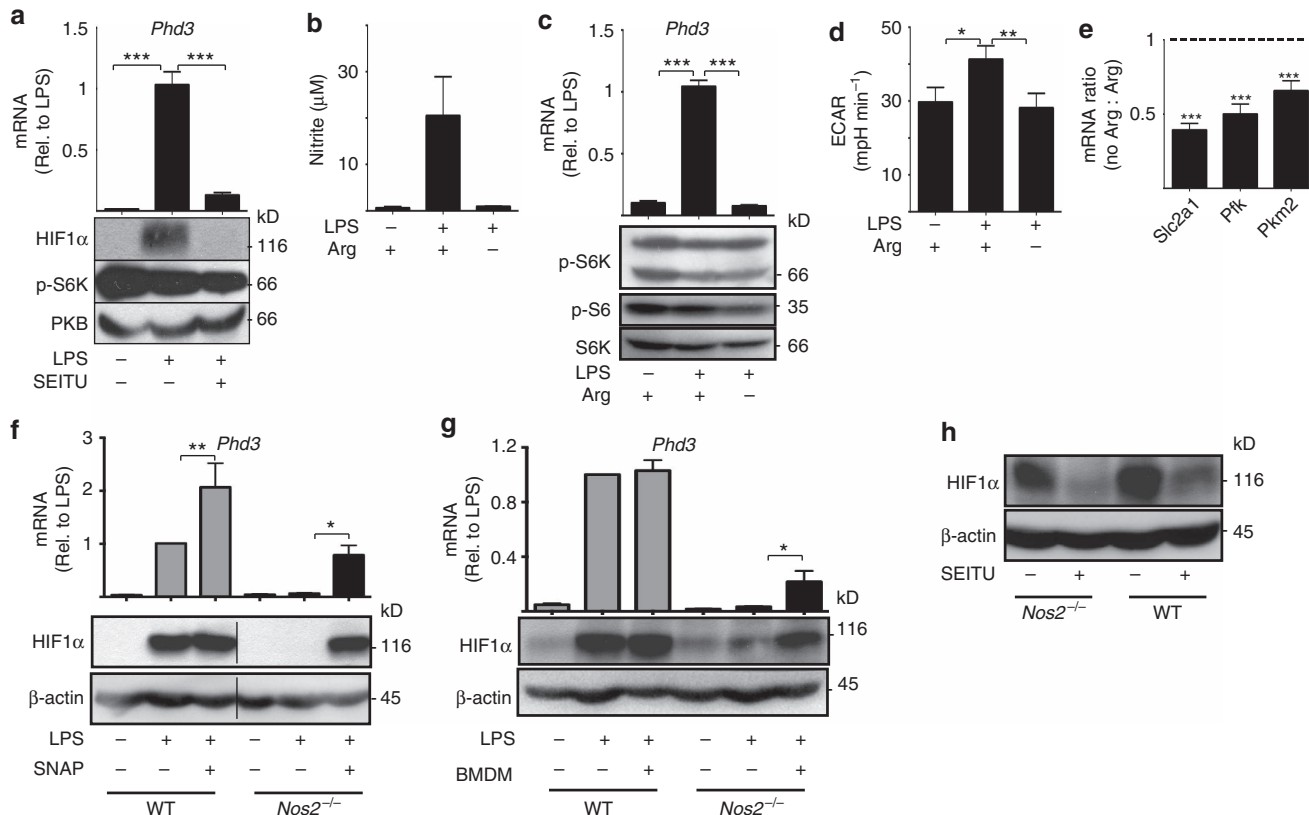

**Figure 6 | iNOS-induced NO is required for HIF1α activity in LPS-activated GM-DCs.** (**a**) GM-DCs were left unstimulated or stimulated with LPS (100 ng ml$^{-1}$) for 20 h in the presence or absence of the iNOS inhibitor SEITU (500 μM), then analysed by qPCR for *Phd3* mRNA expression (upper panel) and by immunoblot analysis (lower panel) for protein levels (phosphorylated p70 S6-kinase, p-S6K; protein kinase B, PKB—loading control). (**b–e**) GM-DCs were left unstimulated or stimulated with LPS (100 ng ml$^{-1}$) for 20 h in the presence or absence of the amino acid arginine (Arg), then analysed (**b**) for nitrite production by the Greiss reaction, (**c**) by qPCR for *Phd3* mRNA expression (upper panel) and by immunoblot analysis (lower panel) for protein levels (phosphorylated S6 ribosomal protein, p-S6; Total p70 S6-kinase, S6K), (**d**) for rates of glycolysis and (**e**) by qPCR for the mRNA expression of the glucose transporter *Slc2a1* and glycolytic enzymes phosphofructokinase (*Pkf*) and Pyruvate kinase 2 (*Pkm2*). (**f,g**) GM-DCs were generated from either wild-type (WT) or iNOS knockout mice (*Nos2$^{-/-}$*), then left unstimulated or stimulated with LPS (100 ng ml$^{-1}$) for 20 h +/− the NO donor S-nitroso-N-acetylpenicillamine (250 μM) (**f**) or +/− BMDMs stimulated with LPS + IFNγ and added to the wells in a transwell insert (**g**) each for the last 4 h of the stimulation. The GM-DCs were then analysed by qPCR for *Phd3* mRNA expression (upper panels) and by immunoblot analysis for HIF1α and β-actin protein levels (lower panels). (**h**) WT and *Nos2$^{-/-}$* GM-DCs were stimulated with LPS (100 ng ml$^{-1}$) for 20 h with LPS + IFNγ-stimulated BMDMs added to the wells in a transwell insert for the last 4 h of the stimulation +/− SEITU (500 μM). Cells were analysed by immunoblot analysis for HIF1α and β-actin protein levels. Data are mean ± s.e.m. at least three separate experiments performed in quadruplicate (**d**) or triplicate (**a–c,e–g**). Representative immunoblots of at least three separate experiments are shown. Data were analysed using a one-way analysis of variance with Tukey's post test (*$P < 0.05$, **$P < 0.01$, ***$P < 0.001$). ECAR, extracellular acidification rate.

increasing the numbers of T cells interacting with the *Nos2$^{-/-}$* GM-DCs resulted in elevated levels of IFNγ production in CD8 T cells (Fig. 8i,j), though the magnitude of the differences were less than those observed in GM-DCs that express iNOS (Fig. 8h, Supplementary Fig. 4d,e).

Next, we investigated whether activating T cells can deprive DCs of nutrients within a lymph node *in vivo*. An experiment was designed to manipulate DC–T-cell interactions within a draining lymph node using an adoptive transfer-based approach. LPS- and SIINFEKL-pulsed GM-DCs were injected into the lateral tarsal region of host mice, and then 4 h later purified CD8 OTI were injected intravenously into the same mice. To manipulate the ratio of T cells to DC, the number of OTI T cells injected was varied. Using this approach, the transferred DC and OTI T cells would arrive at the draining popliteal and inguinal lymph nodes via physiological routes from the tissue and blood, respectively. The draining lymph nodes were harvested, the numbers of transferred GM-DCs and OTI T cells determined and the level of pS6 in the transferred GM-DCs analysed. The ratio of GM-DCs

to OTI T cells in each individual lymph node was determined. In lymph nodes containing <10 OTI T cells per GM-DC, pS6 levels within the GM-DCs were significantly higher than those measured in GM-DCs from lymph nodes where the T-cell–DC ratio was >10 (Fig. 8k,l). These data argue that increased numbers of activating OTI T cells can deprive GM-DCs of glucose *in vivo* to result in the inactivation of mTORC1 signalling. Importantly, the levels of pS6 in activated OTI T cells was not affected by the DC-to-T-cell ratio, indicating that nutrients were not generally limiting within the lymph node (Fig. 8m).

Taken together, these data argue that competitive uptake of glucose is not just a feature of pathological microenvironments but has a role during the induction of normal immune responses to shape the T-cell response.

## Discussion

The present study explores the signalling pathways that link glucose in the local microenvironment to changes in DC

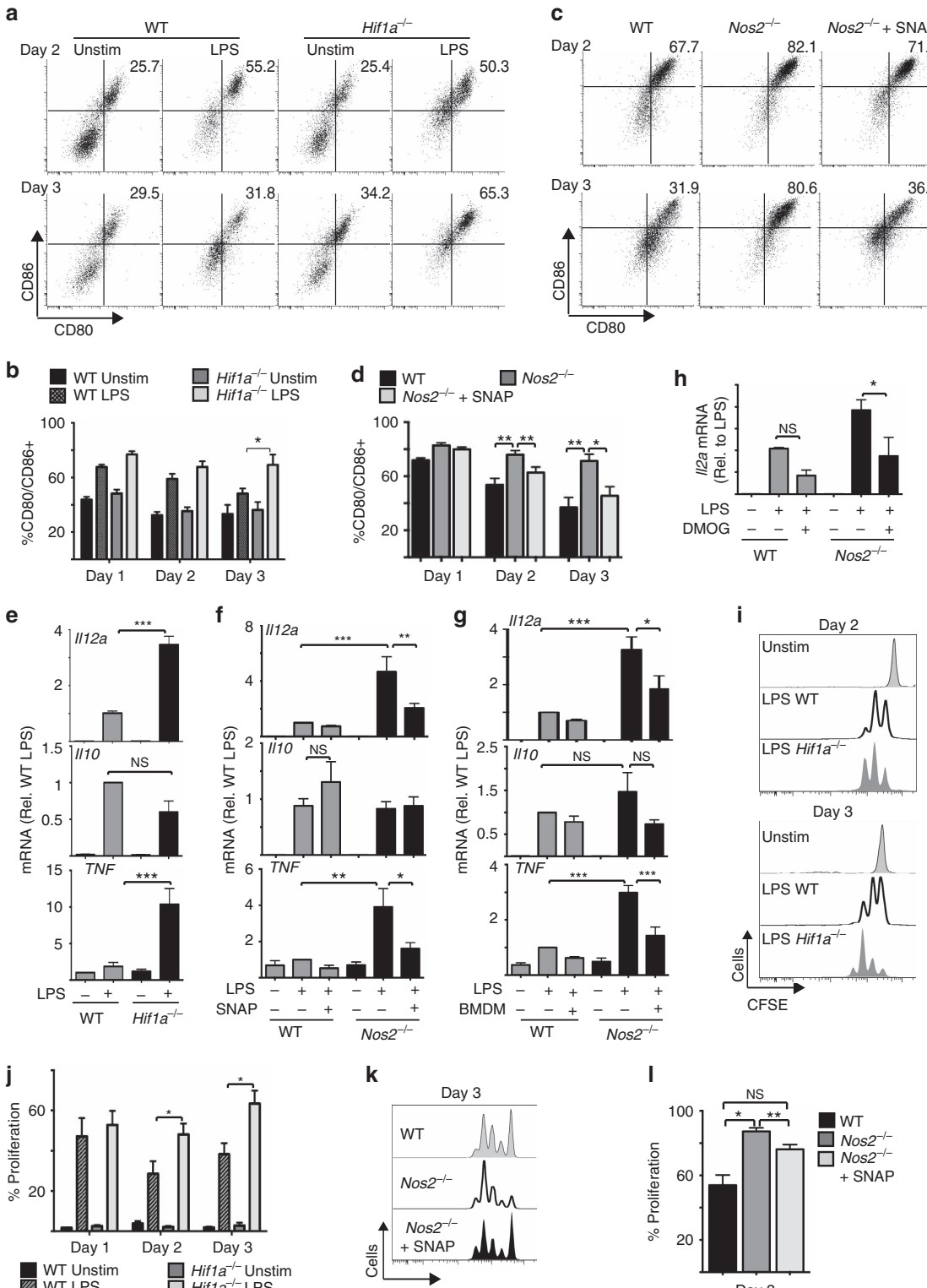

**Figure 7 | LPS-induced iNOS and HIF-1α activity negatively affects GM-DC-induced T-cell responses.** *Hif1a*[flox/flox] (WT) or *Hif1a*[flox/flox] VavCre (*Hif1a*[−/−]) GM-DCs (**a,b,e,i,j**) or WT and *Nos2*[−/−] GM-DCs (**c,d,f–h,k,l**) were pulsed with SIINFEKL peptide +/− LPS (100 ng ml[−1]) and cultured for a 1, 2 or 3 days. *Nos2*[−/−] GM-DCs were treated with S-nitroso-*N*-acetylpenicillamine (500 μM) every 5 h (**c,d**) or for the last 4 h of stimulation (**f**), cocultured with LPS + IFNγ-activated BMDMs (in a transwell cassette) for the last 4 h of stimulation (**g**) or treated with DMOG (200 μM) for the last 4 h of stimulation (**h**). (**a–d**) CD80 and CD86 expression was analysed by flow cytometry and (**e–h**) *Il12a*, *Il10* and *TNF* mRNA by qPCR. (**i–l**) On the indicated day post LPS stimulation, GM-DCs were washed before the addition of purified carboxyfluorescein succinimidyl ester (CFSE)-labelled OT-I T cells. After a co-culture period of 48 h, the OT-I T cells were analysed by flow cytometry for proliferation as measured by CFSE dilution. Data are mean ± s.e.m. or representative of three (**e**), four (**i,j**), five (**a,b,g**), six (**f**) or seven (**c,d,k,l**) separate experiments. qPCR analysis was performed in triplicate. Data were analysed using a one-way analysis of variance with Tukey's post test (*P < 0.05, **P < 0.01, ***P < 0.001).

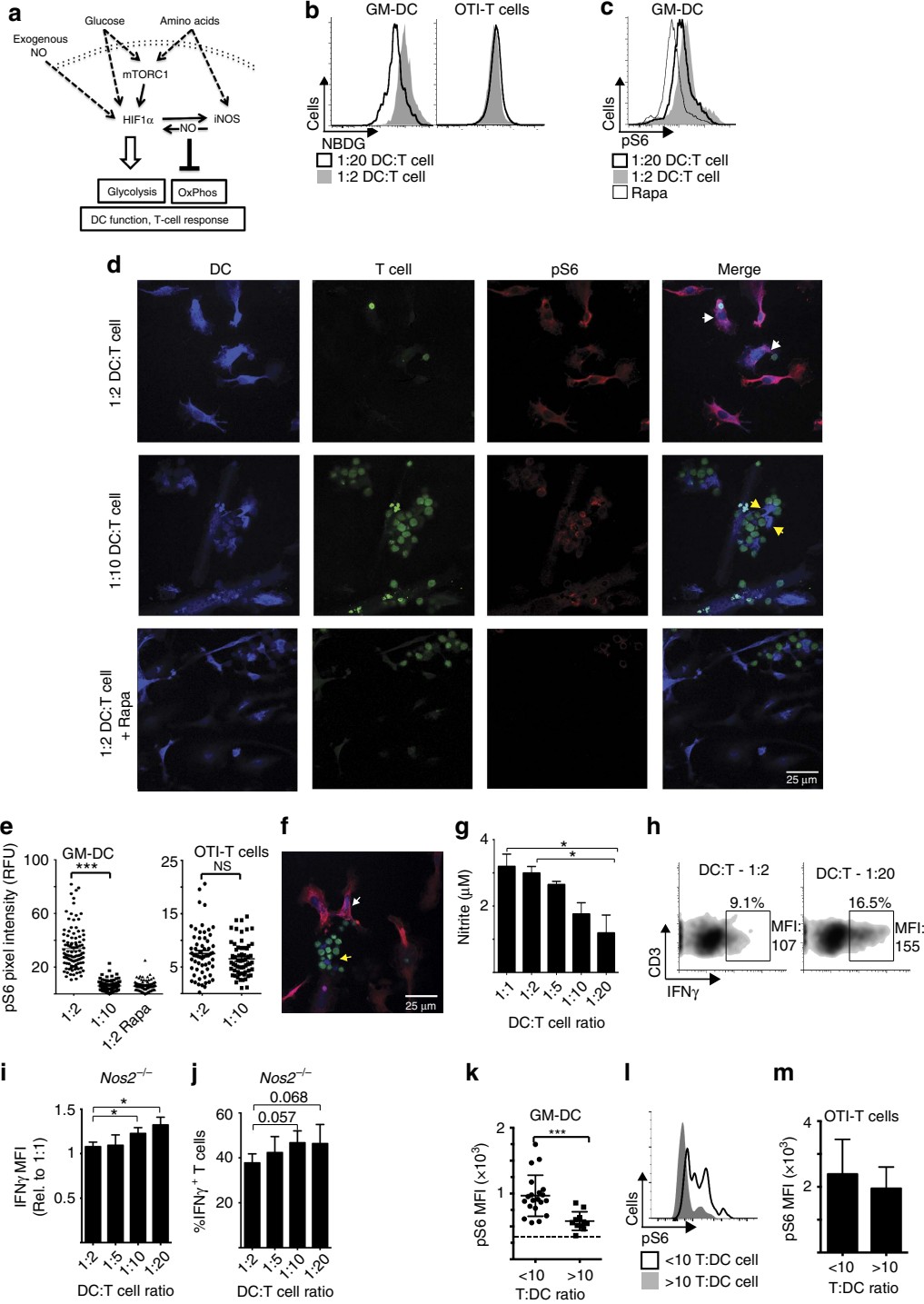

**Figure 8 | Activated T cells deprive GM-DCs of nutrients to alter GM-DC signalling and function.** (**a**) Schematic detailing the mTORC1/HIF1α/iNOS signalling circuit. (**b–j**) GM-DCs were pulsed with SIINFEKL peptide for 6 h +/− LPS (100 ng ml[−1]), washed and purified OT-I T cells were added at different T:DC ratios for 18 h, as indicated. (**b**) NBDG uptake into GM-DCs and CD8 T cells and (**c**) levels of phosphorylated S6 ribosomal protein (p-S6) were measured by flow cytometry. (**d–f**) p-S6 levels were measured by confocal microscopy. GM-DCs were stained with cell tracker violet and T cells with carboxyfluorescein succinimidyl ester (CFSE). Cocultures were treated with rapamycin (Rapa, 20 nM) for the final hour to provide a pS6-negative control (**c–e**). Representative images (**d,f**) and pooled data (**e**) are shown. (**g**) Nitrite production was measured by the Greiss reaction and (**h**) IFNγ production by intracellular flow cytometry. (**i,j**) IFNγ expression in activated CD8 T cells was measured in cocultures using *Nos*[−/−] GM-DCs. Shown is (**i**) IFNγ MFI and (**j**) IFNγ-positive CD8 T cells. (**k–m**) To analyse DC–T-cell interactions *in vivo*, GM-DCs were pulsed with LPS and SIINFEKL peptide for 1 h, washed, stained with cell tracker violet and injected subcutaneously into the lateral tarsal region of C57/B6 host mice. Four hours later, CFSE-stained purified CD8 OT-I T cells were injected intravenously into the same mice ranging from 5 × 10[6] to 2.5 × 10[5] T cells per mouse. Host mice were killed 18 h later and popliteal and inguinal lymph nodes were isolated. The ratio of injected T cells to GM-DCs was determined and pS6 levels were analysed by flow cytometry in the transferred GM-DCs (**k,l**) and CD8 OT-I T cells (**m**). Data are mean ± s.e.m. or representative of three to five separate experiments (**b–j**). Data are mean ± s.e.m. or representative of results for at least 10 separate lymph nodes in each group, from 16 host mice, data were obtained from two separate experiments (**k–m**). Data were analysed using a one-way analysis of variance with Tukey's post test (**b–j**) or Student's *t*-test (**k,m**) (*$P < 0.05$, **$P < 0.01$). MFI, mean fluorescent intensity.

metabolism and function. A key finding was that glucose represses the proinflammatory functions of GM-DCs, inhibiting the induction of T-cell proliferation and IFNγ production. Glucose does not prevent DC-induced T-cell activation but it changes the course of the T-cell response. Glucose-deprived GM-DCs show increased costimulatory molecule and IL12 expression, signals known to be important for the induction of T-cell proliferation and the acquisition of T-cell effector functions[31,32]. Therefore, the observed changes in the T-cell response correlate to the functional changes in glucose-starved GM-DCs. These data argue that glucose represents an important signal that can impact upon the outcome of the T-cell response. This study shows that a complex signalling circuit involving mTORC1, HIF1α and iNOS relays signals from the local DCs microenvironment regarding glucose availability to coordinate the metabolic and functional changes in LPS-stimulated DCs. This circuit is also sensitive to the levels of amino acids such as leucine and arginine that are required for mTORC1 and iNOS activity, respectively. NO that originates from exogenous sources such as local macrophages will also regulate this signalling circuit[33]. Numerous studies have demonstrated that the mTORC1 inhibitor rapamycin increases the proinflammatory outputs of DCs: increased IL12 and costimulatory molecule expression[27–30]. Indeed, inhibition of mTORC1 in human or mouse DCs during TLR stimulation augments the proliferation of effector CD4+ T cells in vitro[29,34,35]. All these studies inhibited mTORC1 pharmacologically, but we now present a model where this can occur physiologically due to nutrient deprivation. Importantly, the data show that nutrient deprivation of Nos2−/− GM-DCs also enhances CD8 T-cell priming, indicating that this model is likely to be applicable to multiple DC subsets. Taken together, this represents a new regulatory axis for the control of DC proinflammatory functions and T-cell responses.

iNOS and mTORC1 have previously been shown to be important for the metabolic shift from OXPHOS to glycolysis in DCs[13,17]. This study now reveals the transcription factor HIF1α as the key molecular link required for the sustained induction of glycolysis in LPS-stimulated DCs. The data show that HIF1α and iNOS activities are closely connected in LPS-activated GM-DCs with the expression of each reliant on the activity of the other. In non-immune cells, hypoxia-induced HIF1α has been shown to bind to DNA elements in the Nos2 promoter and increase gene expression, while NO has been reported to induce HIF1α protein expression, though the exact mechanisms involved are not clear[36–38]. While this signalling circuit will certainly be different in DC subsets that do not express iNOS, it is likely that HIF1α will still be central to the control of cellular glycolysis. Indeed, splenic DCs stimulated in vivo in mice following poly(I:C) injection increase glycolysis and inactivate OXPHOS in a HIF1α-dependant manner even though they do not express iNOS[39]. The question arises as to how these cells inhibit OXPHOS in the absence of iNOS expression. Herein we demonstrate that HIF1α protein expression is induced in GM-DCs that lack the expression of iNOS in response to exogenously derived NO from sources such as proinflammatory macrophages. Additionally, another study demonstrated that iNOS knockout GM-DCs can inhibit OXPHOS normally in response to LPS but only when co-cultured with wild-type GM-DCs, thus demonstrating that NO-dependent inhibition of OXPHOS can be a cell-extrinsic effect[17]. Therefore, it seems likely that exogenously derived NO has an important regulatory role for DC metabolism and function. Certainly, NO produced by phagocytes has been shown to diffuse across membranes to act upon multiple cells in the local microenvironment[33]. Therefore, DC expression of HIF1α appears to be essential for the metabolic

changes that occur in activated DCs, while NO can originate from iNOS expressed within the DCs or may originate from other cells in the local microenvironment.

This study reveals that in GM-DCs, there are two glucose-sensing mechanisms. The AMPK/mTORC1 signalling axis can sense decreasing glucose concentrations to result in the loss of HIF1α protein expression and activity and leading to decreased iNOS expression and NO production. This finding is consistent with our previous work in CD8 T cells that showed glucose withdrawal inhibits mTORC1 signalling following the activation of AMPK[26]. Additionally, the data show that HIF1α protein expression is directly sensitive to glucose deprivation in GM-DCs independently of mTORC1 activity. Recently, it was demonstrated that glucose is required for the protein expression of cMyc in CD8 T cells because glucose feeds the glucosamine pathway that is required for GlcNAcylation of proteins, the reversible addition of UDP-GlcNAc to serine or threonine residues[40]. Therefore, one potential mTORC1-independent mechanism for promoting HIF1α stabilization is through protein GlcNAcylation. Indeed, GlcNAcylation has been linked to HIF1α protein expression by one study in tumour cells[41]. Further work is required to identify mTORC1-independent mechanisms linking glucose availability to HIF1α expression in DCs.

Multiple lines of evidence have described cellular glycolysis as having proinflammatory functions in immune cells[24,42,43]. However, a number of studies have described examples where glycolysis can have immunosuppressive functions; glycolysis is important for certain regulatory T cells subsets both in mice and humans[44,45]; glycolysis is required for the maintenance of DC tolerance in a vitamin D-induced DC tolerance model[46]. Additionally, it is becoming clear that HIF1α can also promote anti-inflammatory effects under certain conditions. For example, under conditions of hypoxia HIF1α has been directly linked to the expression of the immunosuppressive molecules PD-L1 and miR-210 in myeloid cells[47,48]. While most studies to date in T cells and macrophages suggest that HIF1α plays a predominantly proinflammatory role, the data in DCs are less clear. Several papers have shown that HIF1α is required for DC migration, particularly in a hypoxic environment and for IFN production[49–51]. However, an understanding of the role HIF1α plays in DC-induced T-cell responses is still developing. Early reports suggested that HIF1α was required for costimulatory molecule and proinflammatory molecule expression in DCs, as knock down of HIF1α reduced their expression in a hypoxic environment[52]. In contrast, a report suggests that hypoxia affects costimulatory molecules and cytokine production in DCs independently of HIF1α[49]. Overall, the available data support multiple roles for HIF1α in DCs that may differ dependent on the presence or absence of oxygen. More recently, an elegant study of DCs function during Leishmania infection found that HIF1α in DCs promoted a regulatory T-cell response and deleting HIF1α resulted in enhanced DC-dependent IL12 production and increased CD8 T-cell proliferation[53]. This study is consistent with the data presented herein showing an anti-inflammatory role for normoxic HIF1α in DCs to limit the induction of CD8 T-cell responses.

While competitive glucose uptake in pathological situations such as within tumours is now becoming established as a mechanism utilized by tumour cells to alter the course of immune response, the data presented in this study argue that competitive glucose uptake is also a feature of normal immune responses. In fact, in addition to glucose, several amino acids will impact upon mTORC1/HIF1α/iNOS signalling including leucine and glutamine, which are important for mTORC1 activity, and arginine, the fuel for iNOS dependent NO production[7]. The data

generated using both *in vitro* and *in vivo* approaches demonstrates that activating T cells interacting with the antigen-presenting DC can deplete nutrients from the immediate DC microenvironment resulting in the inhibition of this nutrient-sensitive mTORC1/HIF1α/iNOS signalling circuit. During a robust immune response, DC present multiple T-cell antigens and can encourage multiple simultaneous interactions with T cells resulting in the formation of DC–T-cell clusters of up to 12 T cells[21,54]. Given that activated T cells dramatically upregulate rates of glucose and amino acid uptake[5,7], it is perhaps unsurprising that the immediate microenvironment surrounding a DC in such a T-cell cluster becomes nutrient deprived. Therefore, competition for glucose or amino acids will allow these closely interacting cells to adopt contrasting signalling and metabolic states, glycolytic T cells compared to non-glycolytic DCs engaging in oxidative metabolism; both metabolic states that maximize the proinflammatory functions of the respective immune subset. This innovative mechanism allows T cells to promote the proinflammatory functions of the antigen-presenting DC to enhance and prolong the T cell response.

## Methods

**Mice.** Male C57BL/6J mice were purchased from Harlan (Bicester, UK) and maintained in compliance with EU regulations. Permission to perform mouse experiments was granted by the Animal Research Ethics Committee (AREC), Trinity College Dublin and the Health Products Regulatory Authority (HPRA), Ireland. Bones from Hif1a^flox/flox VavCre mice and Hif1a^WT/WT VavCre mice (males and females) were obtained from the Cantrell laboratory in the University of Dundee. OTI transgenic mice were initially purchased from Harlan (Bicester, UK) and then bred in house. Male Nos2 knockout mice were imported from The Jackson Laboratories. All mice were on the C57BL/6J genetic background and were used between the ages of 6 and 20 weeks.

**DC culture.** Bone marrow derived DCs (GM-DC) were generated by culturing bone marrow-derived haematopoietic cells in the presence of 20 ng ml$^{-1}$ GM-CSF for 10 days[55]. Bone marrow was isolated from the femur and tibia-fibula bones of C57/BLJ male mice, followed by RBC lysis. The haematopoietic cells were then counted and cultured at a concentration of 0.4–0.6 × 10$^6$ million cells per ml. The cells were supplemented with 20 ng ml$^{-1}$ GM-CSF (PeproTech) on days 1, 3, 6 and 7 in the course of the 10-day culture. GM-DCs were then cultured in normal RPMI containing 10% fetal bovine serum and 1% Pen/Strep and plated at a concentration of 1 × 10$^6$ cells ml$^{-1}$, unless otherwise stated in the presence of 2 ng ml$^{-1}$ GM-CSF, and stimulated with 100 ng ml$^{-1}$ LPS (Enzo Life Sciences serotype R515 TLRgrade) in the presence of the following compounds where indicated: rapamycin (20 nM), DMOG (200 μM) (Cambridge Bioscience), S-nitroso-N-acetylpenicillamine (250 μM) (Sigma), and SEITU (500 μM) (Cayman Chemical). For glucose-deprivation assays, DCs were stimulated for 8 h in normal RPMI, washed with glucose free RPMI and cultured in glucose-free RPMI supplemented with 1× concentration of MEM Vitamin Cocktail, 1× concentration of selenium/insulin/transferrin Cocktail (Invitrogen/Biosciences) and 10% dialysed FCS (Fisher). Glucose and/or galactose was also added as indicated. Nitrite levels in the GM-DCs culture media was determined using the Griess reaction (Promega, Cat#: G2930).

**BMDM culture.** BMDMs were differentiated from bone marrow-derived haematopoietic cells. Bone marrow was isolated from the femur and tibia-fibula bones of C57/BLJ male mice, followed by RBC lysis. Bone marrow-derived haematopoietic cells were then cultured in 10 cm Petri dishes at a concentration of 1 × 10$^6$ cells ml$^{-1}$ in a volume of 10 ml in DMEM supplemented with 10% FCS, 1% penicillin/streptomycin for 6–9 days. The media was supplemented with 20% of supernatant from the M-CSF-secreting L929 mouse fibroblast cell line.

**BMDM/GM-DC transwell co-culture.** For the co-culture experiments, WT or Nos2$^{-/-}$ GM-DCs were plated at a concentration of 2 × 10$^6$ cells in the presence of 2 ng ml$^{-1}$ GM-CSF and stimulated with 100 ng ml$^{-1}$ LPS for 20 h before analysis. WT BMDMs were cultured at a concentration of 1 × 10$^6$ ml$^{-1}$ in Transwells (Corning Inc.) and were stimulated with 100 ng ml$^{-1}$ LPS and 10 ng ml$^{-1}$ IFNγ. After 12 h of stimulation, BMDMs were treated with another dose of IFNγ (10 ng ml$^{-1}$). The transwells were then transferred into the wells containing GM-DCs. GM-DCs and BMDMs were co-cultured for 4 h before the GM-DCs were lysed for analysis.

**T-cell responses.** DCs were stimulated with LPS (100 ng ml$^{-1}$) and pulsed with OVA 257–264 (SIINFEKL) 1 μg ml$^{-1}$ as indicated and washed extensively in normal media before the addition of magnetic bead (Miltenyi biotech) sorted CD8 OTI T cells. T cells were added at a 5:1 T-cell/DC ratio unless indicated otherwise. T-cell responses were analysed 48 h after addition to GM-DCs unless stated otherwise. Proliferation of T cells was analysed by carboxyfluorescein succinimidyl ester dilution. T cells were treated with phorbol 12-myristate 13-actetate 20 ng ml$^{-1}$ and ionomycin 1 μg ml$^{-1}$ml for 4 h before being fixed and permeabilized for intercellular staining of IFNγ. For 2-NBDG uptake in T-cell/DC co-culture experiments 2-NBDG was added directly to the co-culture wells for the last hour of the experiment, at a final concentration of 35 μM. Cells were then removed from wells, washed extensively and analysed by FACS following surface staining.

**Quantitative real-time PCR.** RNA was extracted using the RNeasy RNA Purification Kit (QIAGEN, cat #: 74106). Purified RNA was reverse transcribed using qScript cDNA synthesis kit (Quanta Biosciences, cat #: 95047). Real-time PCR was performed in triplicate using iQ SYBR Green-based detection on a ABI 7900HT fast qPCR machine. The derived mRNA levels were normalized using RpLp0 mRNA levels. For primer sequences, see Supplementary Table 1.

**Metabolic flux analysis.** Mature BMDCs were plated (4 × 10$^5$ cells per well) in Seahorse culture plate (precoated with Poly-L-Lysine to adhere cells) in 0.25 ml culture media with GM-CSF at 2 ng ml$^{-1}$. After 1 h, a further 0.25 ml culture media was added to wells to stimulate cells with media also containing reagents as required with 2 ng ml$^{-1}$ GM-CSF. After 20 h stimulation, media was removed and replaced with XF media (Seahorse Bioscience) supplemented with GM-CSF (2 ng ml$^{-1}$) and glucose (10 mM). The cell plate was kept at 37 °C for 30 min in a non-CO2 maintaining incubator before insertion into the Seahorse XFe24. The Seahorse XFe24 takes measurements of the extracellular acidification rate (ECAR) and the oxygen consumption rate (OCR) every 7.5 min. Over the course of analysis, four inhibitors are added to determine which processes in metabolism are responsible for the ECAR and OCR rates. The inhibitors, added in the listed order, are oligomycin (2 μM) (inhibits the F0/F1 ATPase), p-trifluoromethoxy carbonyl cyanide phenyl hydrazine (an uncoupling agent) (500 nM), antimycin A (4 μM) and rotenone (100 nM) (inhibit complex 3 and 1, respectively) and 2-deoxy-D-glucose (30 mM; inhibits glycolysis). Metabolic rates were calculated as follows: OXPHOS—basal OCR minus OCR after the addition of antimycin A/rotenone; OXPHOS coupled to ATP synthesis—basal OCR minus OCR after the addition of oligomycin; glycolysis—basal ECAR minus ECAR after the addition of 2DG.

***In vivo* DC–T-cell interaction.** GM-DCs were stimulated for 1 h with LPS (100 ng ml$^{-1}$) and SIINFEKL OVA peptide (1 μg ml$^{-1}$), washed extensively and stained with Cell Tracker Violet (Biosciences Life Technologies). In all, 1 × 10$^6$ GM-DCs (optimized to give 30–200 GM-DCs in the draining lymph node) were injected into the lateral tarsal region of C57/Bl6 mice. Four hours later, different numbers of purified, carboxyfluorescein succinimidyl ester-stained CD8 OTI T cells (1 × 10$^5$–5 × 10$^6$) were introduced by intravenous injection. After 18 h draining, popliteal and inguinal lymph nodes were harvested. Lymph node cells were fixed and analysed for the levels of pS6 in transferred GM-DC and OTI T cells by flow cytometry.

**Immunoblot analysis.** Cells were lysed at 1 × 10$^7$ ml$^{-1}$ in Tris lysis buffer containing 10 mM Tris pH 7.05, 50 mM NaCl, 30 mM Na pyrophosphate, 50 mM NaF, 5 μM ZnCl$^2$, 10% (v/v) Glycerol, 0.5% (v/v) Triton, 1 μM dithiothreitol and protease inhibitors. Lysates were separated by SDS–polyacrylamide gel electrophoresis and transferred to nitrocellulose membrane. Blots were probed with the following antibodies for 4 h at room temperature or overnight at 4 °C: phospho-S6 ribosomal protein$^{S235/236}$ (dilution 1:5,000), phospho-S6K$^{T389}$ (108D2), total S6K, Total PKB(11E7) (1:2,000), phospho-Acetyl-CoA Carboxylase$^{S79}$, total Acetyl-CoA Carboxylase (dilution 1:1,000 Cell Signaling Technology), NOS2 (C-11, dilution 1:200 Santa Cruz Biotech), HIF-1α (dilution 1:2,500; Novus Technologies), and SMC1 (dilution 1:5,000 Bethyl Laboratories) and then incubated with horseradish peroxidase-conjugated anti-rabbit IgG or anti-mouse IgG for 1 h at room temperature. Original immunoblot scans can be seen in Supplementary Fig. 7.

**Flow cytometry.** Cells were labelled with allophycocyanin (APC) CD11c (HL3), BV421 CD11c (HL3), FITC CD80 (16-10A1), CD86 PE (GL1), APC CD40 (1C10), PerCP-eFluor 710 CD40 (1C10), PE-Cy7 CD19 (1D3), PerCP-Cy5.5 (H1.2F3), APC TCRβ (H57-597), phycoerythrin (PE) TCRβ (H57-597), FITC CD3 (145-2c11), APC-eFluor 780 major histocompatibility complex II (M5/114.15.2), BV605 CD45.1 (A20), BV786 CD45.2 (104) and APC IFNγ (XMG1.2) purchased from eBioscience or BD Pharmingen. pS6 analysis used phospho-S6 Ser 235/236 (Cell Signaling Technologies) and secondary was PE-conjugated donkey anti-rabbit immunoglobulin G (Jackson ImmunoResearch). 2-NBDG (Life Technologies) was added to cells at 35 μM for 1 h prior to analysis. Live cells were gated by forward

scatter (FSC-A) and side scatter (SSC-A) analysis. Single cells were selected by FSC-A and FSC-W analysis. For intracellular staining, cells were then fixed and permeabilized using Cytofix/Cytoperm reagent (BD Pharmingen). For cytokine analysis, endocytosis was blocked using golgi plug (BD Pharmingen) for 4 h. Gating strategies for all flow cytometry analysis are outlined in Supplementary Figs 5 and 6. Data were acquired on a FACSCanto or LSRFortessa (Becton Dickinson) and analysed using the FlowJo software (TreeStar).

**Confocal microscopy.** Cell tracker Violet (Thermo fisher scientific) stained BMDCs were plated overnight on concentrated nitric acid-treated glass coverslips. DCs were stimulated with LPS (100 ng ml$^{-1}$), pulsed with OVA 257-264 (SIINFEKL, 1 µg ml$^{-1}$) for 6 h, washed extensively and purified CD8 OTI T cells added for 18 h at a 1:10 or 1:2 DC:T-cell ratio before fixing in 4% paraformaldehyde. Cells were blocked/permeabilized for 1 h (1% serum/0.3% Triton-X), incubated for 2 h in anti-phospho-S6 ribosomal protein$^{S235/236}$ (Cell Signalling, dilution 1:100) washed and then with anti-rabbit Alexa Fluor 594 secondary antibody (dilution 1:500; Invitrogen/Molecular Probes) for 1 h. The coverslips were mounted with Hydromount (National Diagonostics) and immunofluorescence images were captured using a Leica SP8 gate STED confocal microscope. Images were acquired and quantified using the Leica LAX software.

**Statistics.** Data were analysed using Graphpad Prism version 6.01 for Macintosh (Graphpad Software). Data are presented in graphs as mean ± s.e.m. Multiple comparisons for large groups used a one-way analysis of variance where variances were considered equal, with Tukey's posttest to compare individual groups. Student's $t$-tests were used when comparing two data sets.

**Data availability.** The authors declare that the data supporting the findings of this study are available within the article and its Supplementary Information files.

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

## Acknowledgements

This work was supported by funding from Science Foundation Ireland (13/CDA/2161) and Marie Skłodowska-Curie Actions (PCIG11-GA-2012-321603). S.J.L. was supported by a MolCellBio PhD scholarship funded by the Programme for Research in Third-Level Institutions (PRTLI). J.F.W was supported by a Wellcome Trust Studentship (106811/Z/15/Z).

## Author contributions

D.K.F. and S.J.L. conceptualized the project; D.K.F., S.J.L., N.K.-M., R.M., J.F.W. and O.C. performed the experiments; D.K.F. wrote the original draft of the manuscript; D.K.F., J.M., S.J.L. and N.K.-M. reviewed and edited the manuscript. L.V.S. and M.N.N. provided resources essential to this study. D.K.F. and J.M. supervised the research.

## Additional information

**Competing interests:** The authors declare no competing financial interests.

