## [Peer Review File · Nature Communications]

Reviewers' comments:

Reviewer #1 (Remarks to the Author):

In the current manuscript Lawless et al. describe how glucose through a mTOR/HIF1a/iNOS dependent pathway limits Dendritic cell activation and their T cell priming capacity. As consequence, their data may suggest that T cells interacting with DCs compete for glucose thereby promoting the T cell activating capacity of these APCs. While the study is generally well executed, the manuscript well written, and these findings can be potentially of great interest to immunologists and to the field of immunometabolism, I feel some of the current conclusions are still premature and I have concerns about the appropriateness of the DC model used in this study. I have the following comments that need to be addressed before it can be considered for publication.

Major comments:

1) The authors use GM-CSF DCs to study the role of glucose in DC activation and T cell priming ability. While GM-DCs are a great tool to study many aspects of general DC biology, they may be metabolically very differently wired than conventional DCs because in contrast to cDCs they express iNOS in response to activation. This is relevant from a metabolic standpoint because iNOS expression has such profound effects on the metabolism of these cells: blocking mitochondrial respiration in a NO dependent manner (Everts et al 2012)). The current observations in GM-DCs (more activation of DCs and T cell priming by DCs in low glucose) are largely dependent on iNOS expression. Therefore, it remains to be seen whether they will be applicable to cDCs, the type of DCs that are generally considered to prime T cells. As such it remains to be seen whether a similar effect is observed in DCs that do not express iNOS in response to activation and how generally applicable the findings are. Thus, It will be crucial to repeat some of the key findings from this study (glucose-mtor/Hif1 axis, T cell priming in the absence of glucose etc) with conventional DCs or NOS2 KO GM-DCs at the minimum.

2) The authors replace glucose by galactose to study the role of glucose in DC activation and T cell priming. This is valid approach but it would important to see what happens in the absence of a sugar source or when glycolysis is inhibited (using 2DG for instance) with respect to DC activation, Oxphos, HIF1 expression, iNOS expression and T cell priming. This would help to tell whether some of the effects are due to the absence of glucose or due to the presence of galactose.

3) Fig1 /2

a. In figure 1f only IL12a is analyzed. Why do the authors focus on IL12a specifically? The authors should analyze other cytokines as well (IL6, TNF, IL1), preferably at protein level. And is IL-10 reversely affected?

b. Is enhanced T cell priming by the galactose treated DCs also evident when the cells are fed with OVA protein instead of only the peptide? In other words does glucose reduce antigenprocessing as well?

4) Fig 3

a. The authors find that that lowering glucose levels activated AMPK while galactose treatment does not. The authors suggest that this may be due enhanced oxphos specifically in the presence of galactose (fig 3h). Why wouldn't the metabolic response occur in low glucose levels? The authors should test oxphos in low glucose levels.

b. To explain why in glucose-treated DC HIF1a is expressed/stabilized but not in galactose treated cells, the authors suggest that perhaps this is due to the role that glucose may play in O-glcNAcylation of HIF1a. While this is potentially a very interesting angle, the conclusions that are drawn from the presented data are very premature and not (yet) supported by the data . First, using an OGT inhibitor, affects global O-glcNAcylation- it does not prove that O-glcNAcylation of

HIF1a itself is important for HIF1a stability. It may act indirectly. Second, using an OGT inhibitor does not prove a role for glucose per se in O-glcNAcylation. To really link glucose to HIF1a stability via O-glcNAcylation one would need to assess HIF1a O-glcNAcylation sites in the presence and absence of glucose and galactose.

5) Fig 6

a. The authors interrogate the role of iNOS activity HIF1a expression by using various approaches including treatment with SNAP and SEITU and culturing the cells in the absence of presence of arginine. While the first two give clear results, the last one may also work indirectly: Arginine deprivation may result in loss of HIF1 due to lower mTORC activity. The authors should assess mTORC targets phosphorylation in the presence or absence of arginine to test this.

6) Fig 7

a. In fig 7c/d/e IL12a is analyzed. Can the authors comment on whether other cytokines show similar trends?

b. In figure 7f only IL12p70 is analyzed. Why do the authors focus on IL12 specifically? The authors should analyze some other markers of DC activation as well to determine how general the effect is.

7) Figure 8. The data presented in figure 8 are very interesting, but I think the data are open for multiple interpretations that should be addressed by the authors.

a. The authors say that their data argue that T cell can limit glucose uptake into DCs due to competition. However, no data is shown that really shows its competition for glucose. T cells are sensitive to glucose competition (Chang et al, 2013/2015), so why would only DCs (8b) be affected and not T cells (8c)? This may argue that it's not local glucose deprivation. I could think of two alternatives:

i. Wouldn't it be possible that the immunological synapse between DC and T cell somehow lead to a signal into the DCs to lower glucose uptake (leaving T cells unaffected)? For instance what happens to glucose uptake LPS activated DCs that are treated with agonistic aCD40 or trimeric CD40 Ligand?

ii. Could it be that arginine is deprived by T cells, (directly explaining fig 8g) thereby lowering mTOR activation, HIF1a expression and glucose uptake. In this scenario drop in glucose uptake is not a driving factor, but the consequence. What happens to the data presented in fig 8 if a surplus of arginine is added to the media?

b. The data shown in figure 8d are superfluous as similar data are shown in fig 3e.

c. The interpretation of the in vivo data (fig j/k/l) would be greatly helped if 2NBDG data of DCs and T cells are shown to be able to link it better to the in vitro data. Also the data presented in these panels should be based on number of CD69+ T cells to be able to provide a better correlate for the T cells that have recently interacted or are interacting with the DCs.

d. How does NO production by DCs affect T cell priming? One could imagine that NO may negatively affect T cell activation/proliferation. Could the authors comment on how this would play into their system.

Minor comments:

1) On page 7 the authors state that 'fig 3 links HIF1a to LPS-induced glycolytic reprogramming in GM-DCs'. That's not true - this would be a valid conclusion only after figure 4.

2) Also on page 7 the authors say: 'these data indicate that HIF1a is the key glycolytic regulator that is required for LPS-induced glycolytic reprogramming'. The authors should be a little more careful with this statement as it was previously shown that early glycolytic reprogramming in DCs (before iNOS/HIF1a axis comes into play) is HIF1a independent (Everts et al 2014). So adding 'longterm' to this phrase would suffice.

3) There were some recent papers from Jonathan Powell's and Doug Green's group showing asymmetric metabolic division of T cells when interacting with APCs. The authors should speculate whether their current findings could play a role in this process.

Reviewer #2 (Remarks to the Author):

Overall, this is a well-executed study using a series of in vitro experiments to show a role for T cells in competing with DCs for glucose. The authors found that this competition is important for regulation of antigen-stimulated DCs by negatively regulating DC-induced T cell responses. The authors also simulate a physiological response to argue that a similar condition of glucose competition occurs in vivo.

1) There is some over-reliance on pharmacological inhibitors, ST045849, SIETU, and DMOG. These should be substantiated with some genetic evidence (e.g. VHL $-/-$ T cells). The link between O-GlcNAC to T cell activation via DCs needs to be formally demonstrated with a functional T cell assay (e.g. CFSE, IFNg). The authors should also show at the concentrations of the ST045849 that O-GlcNAC is indeed repressed in DCs.

2) Throughout the manuscript, it is not clear whether other metabolites are also in competition. Can the authors comment on this point? How much does the glucose concentration in the media change? In Fig. 6, the authors should show that arginase in the T cell competition assay is also depleted in the GM-DC co-cultures.

3) Figure 8. The claim that T cell-induced glucose deprivation needs to be shown by measuring 2NBDG and ECAR in DCs and T cells harvested from the LNs. The conclusion that HIF-1 contributes to DC dysfunction should be repeated with HIF-1 k/o OTI cells in this figure. These two are important experiments to substantiate the in vivo conclusions.

4) Some other important references have been omitted:
- Krawczyk et al., as the first description of LPS-induced glycolysis in DCs.
- Wang et al., (PNAS 2013) on TSC1 $-/-$ in DCs

Having said all this, the main concern I have is whether the manuscript represents a sufficient conceptual advance on this topic. Although I acknowledge there are some novel aspects in this study (e.g. formal demonstration of glucose competition by T cells), many of the conclusions have been previously reported (and cited by the authors) by the Pearce group and others (e.g. the role of NO on DC glucose metabolism, contribution of HIF-1). Moreover, is the co-culture with T cells to demonstrate glucose competition different from removal of glucose from the media, as shown by the aforementioned studies?

Minor

1) Page 5; some of the Figure number in this section is incorrect. Please verify and edit as appropriate.

Reviewer #3 Remarks to the Author):

There is an increasing appreciation for the role of metabolic programming in regulating the activation and differentiation of immune cells. This work serves to advance our knowledge by

examining the role of glucose in repressing Dendritic Cell activation and the subsequent T cell responses. Specifically, the group demonstrates the ability of glucose availability to regulate glycolysis through HIF-1a and iNOS. Overall the data are robust and make exciting new connections. It would be great however, if the authors might address the following issues.

Figure 7 was very rewarding in that it took the findings regarding HIF1a, mTOR and iNOS and related them to the initial observations of the paper. However, I think it is crucial to do the same for low glucose levels (not just galactose). Meaning Figure 3d implies that lowering glucose levels achieves the same effect as culturing in galactose. Figure 3h tells us that maybe they aren't exactly equivalent. However, I think it is important that observations along the lines of Figure 7c,d,e, be shown for the low glucose levels

Second, many aspects of Figure 8 cannot really be interpreted as robustly as the authors imply. Co-culturing T cells with DC's will have more effects than just depriving the DC's of glucose. Figure 8b is nice. Could the authors then take those populations and maybe simply show some differential molecular data for the 1:2 and 1:20 DC's?

Point by point response to the reviewers comments (response in red)

Reviewer 1:

1) The authors use GM-CSF DCs to study the role of glucose in DC activation and T cell priming ability. While GM-DCs are a great tool to study many aspects of general DC biology, they may be metabolically very differently wired than conventional DCs because in contrast to cDCs they express iNOS in response to activation. This is relevant from a metabolic standpoint because iNOS expression has such profound effects on the metabolism of these cells: blocking mitochondrial respiration in a NO dependent manner (Everts et al 2012)). The current observations in GM-DCs (more activation of DCs and T cell priming by DCs in low glucose) are largely dependent on iNOS expression. Therefore, it remains to be seen whether they will be applicable to cDCs, the type of DCs that are generally considered to prime T cells. As such it remains to be seen whether a similar effect is observed in DCs that do not express iNOS in response to activation and how generally applicable the findings are. Thus, It will be crucial to repeat some of the key findings from this study (glucose-mTOR/Hif1 axis, T cell priming in the absence of glucose etc) with conventional DCs or NOS2 KO GM-DCs at the minimum.

We agree that it is important to show that our findings are applicable to DC that do not express iNOS. There is clear evidence in the literature that cDC that do not express iNOS inactivate OxPhos in response to TLR signalling as has been described for iNOS expressing BMDC (Everts et al., 2012; Krawczyk et al., 2010; Pantel et al., 2014). This argues that in vivo exogenous sources of NO can impact upon the metabolism of cDC. Therefore, we tested his hypothesis in iNOS KO BMDC and have included the data in the revised manuscript. We show that exogenously supplied NO is sufficient to drive HIF1 α protein expression both by adding the NO donor SNAP and by culturing GM-DC with LPS and IFN γ stimulated BMDM in a transwell system. Addition, of SNAP or BMDM for just the last 4 hours of the 20 hour stimulation was sufficient to

stimulate HIF1 α protein expression and inhibit the expression of PHD3, IL12a.

We have also shown that exogenous NO is sufficient to inhibit the expression of costimulatory molecules on LPS activated Nos2KO GMDC and inhibit Nos2KO GMDC induced T cell proliferation.

This new data argues that our findings are indeed applicable to DC that do not express NOS2 as the NO can be generated by another cell in the local microenvironment.

2) The authors replace glucose by galactose to study the role of glucose in DC activation and T cell priming. This is valid approach but it would important to see what happens in the absence of a sugar source or when glycolysis is inhibited (using 2DG for instance) with respect to DC activation, Oxphos, HIF1 expression, iNOS expression and T cell priming. This would help to tell whether some of the effects are due to the absence of glucose or due to the presence of galactose.

Targeting glycolysis with 2DG is a worthwhile approach and has helped produce some exciting results related to HIF1 in macrophages(Tannahill et al., 2013). However our concern with this was the increasing evidence that 2DG has influences outside of effects on metabolic pathways. It has been shown to inhibit cellular glycosylation and also induce endoplasmic reticulum stress, effecting protein expression(Andresen et al., 2012; Yu and Kim, 2010). We felt glucose replacement with galactose was the better alternative, as cells are not metabolically stressed under these conditions while allowing the specific inhibition of glycolysis. Supplementary figure 1 we think addresses the question if the effects are due to an absence of glucose or presence of galactose. When GMDCs are cultured in both 10mM glucose and 10mM galactose the GMDCs seem to preferentially use glucose and the effects on induced T cell proliferation are nearly identical to GMDCs cultured in glucose alone.

3) Fig1 /2

a. In figure 1f only IL12a is analyzed. Why do the authors focus on IL12a specifically? The authors should analyze other cytokines as well (IL6, TNF, IL1), preferably at protein level. And is IL-10 reversely affected?

b. Is enhanced T cell priming by the galactose treated DCs also evident when the cells are fed with OVA protein instead of only the peptide? In other words does glucose reduce antigen processing as well?

IL12 expression was of particular interest to us as this cytokine has well a described role in the induction CD8 T cell responses, including clonal expansion (Starbeck-Miller et al., 2014; Valenzuela et al., 2002). We have now also analysed IL10 expression and include the data in the revised manuscript.

This is an outstanding question and we will have to analyze this at a future point. However, it should be noted that the GM-DCs were in glucose for the first 8 hours of activation when antigen processing would be occurring and were only switched into galactose at 8 hours after activation.

4) Fig 3

a. The authors find that that lowering glucose levels activated AMPK while galactose treatment does not. The authors suggest that this may be due enhanced oxphos specifically in the presence of galactose (fig 3h). Why wouldn't the metabolic response occur in low glucose levels? The authors should test oxphos in low glucose levels.

b. To explain why in glucose-treated DC HIF1a is expressed/stabilized but not in galactose treated cells, the authors suggest that perhaps this is due to the role that glucose may play in O-glcNAcylation of HIF1a. While this is potentially a very interesting angle, the conclusions that are drawn from the presented data are very premature and not (yet) supported by the data . First, using an OGT

inhibitor, affects global O-glcNAcylation- it does not prove that O-glcNAcylation of HIF1a itself is important for HIF1a stability. It may act indirectly. Second, using an OGT inhibitor does not prove a role for glucose per se in O-glcNAcylation. To really link glucose to HIF1a stability via O-glcNAcylation one would need to assess HIF1a O-glcNAcylation sites in the presence and absence of glucose and galactose.

Lowering glucose levels has been shown to activate AMPK in T cells (Rolf et al., 2013). Although you may get a temporary increase in oxphos in low glucose by way of compensation for the reduced glycolysis, eventually these cells would become energy stressed and switch on AMPK leading to inhibition of mTORC1. Potentially cells could use other carbon sources, e.g. glutamine, to fuel oxphos however GMDCs do not appear to do this. Galactose cultured cells can maintain their ATP pool because they use galactose as a fuel to sustain OxPhos (see figure 5a).

We accept the comments from reviewers 1 and 2 that the o-GlcNAcylation data was somewhat preliminary and needed further validation. We have started to study this in detail using a range of approaches but now feel that this would be more appropriate for a separate manuscript. We have removed the data on O-GlcNAcylation and now simply state that mTORC1 dependent and independent mechanism link glucose to the expression of HIF1a. GlcNAcylation is mentioned in the discussion as a possible mTORC1 independent mechanism.

5) Fig 6

a. The authors interrogate the role of iNOS activity HIF1a expression by using various approaches including treatment with SNAP and SEITU and culturing the cells in the absence or presence of arginine. While the first two give clear

results, the last one may also work indirectly: Arginine deprivation may result in loss of HIF1 due to lower mTORC activity. The authors should assess mTORC targets phosphorylation in the presence or absence of arginine to test this.

In other cell types arginine is required for mTORC1 signaling (Carroll et al., 2016). However, arginine deprivation does not impair mTORC1 signaling in many immune cell subsets including GM-DCs. This data is now included in Figure 6c. We have made similar observations in CD8 T cells and NK cells where arginine is not essential for maintaining mTORC1 activity.

6) Fig 7

a. In fig 7c/d/e IL12a is analyzed. Can the authors comment on whether other cytokines show similar trends?

b. In figure 7f only IL12p70 is analyzed. Why do the authors focus on IL12 specifically? The authors should analyze some other markers of DC activation as well to determine how general the effect is.

We have now also included data for IL10 and TNFalpha for HIF1aKO and Nos2KO GM-DC.

7) Figure 8. The data presented in figure 8 are very interesting, but I think the data are open for multiple interpretations that should be addressed by the authors.

a. The authors say that their data argue that T cell can limit glucose uptake into DCs due to competition. However, no data is shown that really shows it's competition for glucose. T cells are sensitive to glucose competition (Chang et al, 2013/2015), so why would only DCs (8b) be affected and not T cells (8c)? This may argue that it's not local glucose deprivation. I could think of two alternatives:

i. Wouldn't it be possible that the immunological synapse between DC and T cell somehow lead to a signal into the DCs to lower glucose uptake (leaving T cells unaffected)? For instance what happens to glucose uptake LPS activated DCs that are treated with agonistic aCD40 or trimeric CD40 Ligand?

ii. Could it be that arginine is deprived by T cells, (directly explaining fig 8g) thereby lowering mTOR activation, HIF1a expression and glucose uptake. In this scenario drop in glucose uptake is not a driving factor, but the consequence. What happens to the data presented in fig 8 if a surplus of arginine is added to the media?

b. The data shown in figure 8d are superfluous as similar data are shown in fig 3e.

c. The interpretation of the in vivo data (fig j/k/l) would be greatly helped if 2NBDG data of DCs and T cells are shown to be able to link it better to the in vitro data. Also the data presented in these panels should be based on number of CD69+ T cells to be able to provide a better correlate for the T cells that have recently interacted or are interacting with the DCs.

d. How does NO production by DCs affect T cell priming? One could imagine that NO may negatively affect T cell activation/proliferation. Could the authors comment on how this would play into their system.

We accept that we needed to show extra data for our DC:T cell model and attempted to address this by some imaging studies. We can now show that at high T cell:DC ratios T cells will cluster around a single DC, similar to observations by Bousso et al made in intact lymph nodes (Bousso and Robey, 2003). This clustering would allow the T cell to have access to nutrients but the DC, surrounded by T cells, would have limited nutrients available to it. This is evident in the representative figure were the DCs in the 1:10 DC:T cell ratio, surrounded by T cells, have suppressed mTORC1 signaling (Figure 8d). However there were a few DCs in the 1:10 ratio coculture that did not interact with T cells and these DCs had high mTORC1 activity, showing there was no global depletion of nutrients (Figure 8f).

We agree that there is a lot more going on at the DC T cell synapse than just changes in nutrient availability. CD40 ligation increases DC activation but to our knowledge the effects of this event on glucose levels has not been demonstrated. An observation we have that is the subject of future investigations is that at low T cell:DC ratios (e.g. 2:1), when signaling may have an influence independent of nutrient competition, there is a slight increase in both pS6 and glucose uptake (Suppl Figure 4b,c) . This argues that T cell/DC interactions in themselves actually increased mTORC1 signalling and NBDG uptake. However at higher levels when clustering occurs and nutrient will become limiting, there is a significant reduction in both of these measurements.

2NBDG in vivo see comments to reviewer 2 below!

NO is viewed mainly as a pro-inflammatory signal, due to its antimicrobial activity. However it has also been demonstrated to inhibit T cell proliferation(Sato et al., 2007). In our system NO is tightly linked to the glycolytic switch in GMDCs, with any intervention that blocks glycolysis also blocking NO production. In our minds this would suggest that when a DC is interacting with a T cell, there is mechanisms in place that shut down NO production to enhance DC induced T cell responses.

Minor comments:

1) On page 7 the authors state that 'fig 3 links HIF1a to LPS-induced glycolytic reprogramming in GM-DCs'. That's not true - this would be a valid conclusion only after figure 4.

Agreed. This phrase has been changed to “The data in figure 3 suggested that HIF1 α may be required for LPS-induced glycolytic reprogramming in GM-DCs”

2) Also on page 7 the authors say: 'these data indicate that HIF1a is the key glycolytic regulator that is required for LPS-induced glycolytic reprogramming'. The authors should be a little more careful with this statement as it was previously shown that early glycolytic reprogramming in DCs (before iNOS/HIF1a axis comes into play) is HIF1a independent (Everts et al 2014). So adding 'longterm' to this phrase would suffice.

Thank you for point this out. Yes, our statement should have made it clear that we are talking about the glycolytic reprogramming that happens after 12-18 hours. We have amended as suggested.

3) There were some recent paper from Jonathan Powell 's and Doug Green's group showing asymmetric metabolic division of T cells when interacting with APCs. The authors should speculate whether their current findings could play a role in this process.

We are aware of these studies but I do not think that limited nutrient supply is impacting upon the asymmetry in that setting. Their data argues that the altered metabolism in each of the daughter cells is due to unequal sharing of the amino acid transporter slc7a5 that leads one daughter cell to be able to take up leucine and activate mTORC1/Myc signalling and a glycolytic metabolism.

Reviewer #2 (Remarks to the Author):

Overall, this is a well-executed study using a series of in vitro experiments to show a role for T cells in competing with DCs for glucose. The authors found

that this competition is important for regulation of antigen-stimulated DCs by negatively regulating DC-induced T cell responses. The authors also simulate a physiological response to argue that a similar condition of glucose competition occurs in vivo.

1) There is some over-reliance on pharmacological inhibitors, ST045849, SIETU, and DMOG. These should be substantiated with some genetic evidence (e.g. VHL -/- T cells). The link between O-GlcNAC to T cell activation via DCs needs to be formally demonstrated with a functional T cell assay (e.g. CFSE, IFNg). The authors should also show at the concentrations of the ST045849 that O-GlcNAC is indeed repressed in DCs.

We agree that our study could benefit from additional genetic evidence. We have now included data using Nos2 KO mice that correlates with all the data obtained using pharmacological approaches and also with the HIF1a KO GM-DC data.

We accept the comments from reviewers 1 and 2 that the o-GlcNAcylation data was somewhat preliminary and needed further validation. We have started to study this in detail using a range of approaches but now feel that this would be more appropriate for a separate manuscript. We have removed the data on O-GlcNAcylation and now simply state that mTORC1 dependent and independent mechanism link glucose to the expression of HIF1a. GlcNAcylation is mentioned in the discussion as a possible mTORC1 independent mechanism.

2) Throughout the manuscript, it is not clear whether other metabolites are also in competition. Can the authors comment on this point? How much does the glucose concentration in the media change? In Fig. 6, the authors should show that arginase in the T cell competition assay is also depleted in the GM-

DC co-cultures.

We do believe that other metabolites may also be in competition. In particular T cells have high rates of glutamine uptake and leucine uptake. Deprived DC of either of these would also result in the loss of mTORC1 signalling within the DC. These points have now been added to the discussion. Unfortunately the tools analogous to 2NBDG that would be required to measure competitive uptake of nutrients such as glutamine and leucine do not exist yet. We are hoping to try to develop these technologies in the near future to allow us to ask these questions.

If the DC were depleted of arginine there would certainly be a loss of NO production but arginine deprivation does not inactivate mTORC1 in GM-DC. Therefore, arginine depletion is not compatible with the observed decreased in pS6 in GM-DCs.

3) Figure 8. The claim that T cell-induced glucose deprivation needs to be shown by measuring 2NBDG and ECAR in DCs and T cells harvested from the LNs. The conclusion that HIF-1 contributes to DC dysfunction should be repeated with HIF-1 k/o OTI cells in this figure. These two are important experiments to substantiate the in vivo conclusions.

We agree that knowing the glycolytic rates of in DCs and T cells harvested from LNs would be very informative. However, this is technically impossible as it would require purifying DCs from 100s of mice. We have 200-300 DC per popliteal LN and require 250000 per seahorse well.

Looking at 2NBDG in vivo is also very technically challenging primarily due to the relatively weak intensity of fluorescence associated with NBDG. Ed Pearce's group did show increased glucose uptake into LPS stimulated CD11b+ DC in vivo though the difference in MFI is very small (see figure from Everts et

al, 2014 PMID:24562310 below). They were not able to detect increased NBDG in CD8⁺ DC which have less of a glycolytic response

NBDG uptake in vivo is a rather insensitive assay. For our experiments we have the added complication that GM-DC have a higher amount of autofluorescence that fluoresces in the same range as NBDG. We are developing alternate glucose uptake assays that emits in the near infrared spectral region that we believe will allow for glucose uptake to be measured robustly in vivo. But these tools are at an early stage of development. But we hope to be able to address these questions in subsequent studies.

4) Some other important references have been omitted:

- Krawczyk et al., as the first description of LPS-induced glycolysis in DCs.
- Wang et al., (PNAS 2013) on TSC1^{-/-} in DCs

This was an oversight, apologies. These references have now been included.

Having said all this, the main concern I have is whether the manuscript represents a sufficient conceptual advance on this topic. Although I acknowledge there are some novel aspects in this study (e.g. formal demonstration of glucose competition by T cells), many of the conclusions have been previously reported (and cited by the authors) by the Pearce group and others (e.g. the role of NO on DC glucose metabolism, contribution of HIF-1). Moreover, is the co-culture with T cells to demonstrate glucose competition different from removal of glucose from the media, as shown by the aforementioned studies?

We acknowledge the concerns of the reviewer about if this paper represents a sufficient conceptual advance but we hope to convince the reviewer that this paper does add greatly to the immunometabolism field. The individual molecules studied in this paper (e.g. NO, HIF1) have been implicated in immune metabolism before. However our paper is the first to demonstrate comprehensively the connections between all these molecules during GM-DC activation. We now also include data to show that NO can come from external sources such as from macrophages to influence this signalling circuit; NO from co-cultured BMDM promotes HIF1 α expression in Nos2 KO GM-DC. This data provides an explanation for the metabolic changes observed in vivo in cDC (that do not express iNOS) following poly(I:C) injection ((Pantel et al., 2014).

The Pearce group has nicely shown initial metabolic changes in DC that happen minutes after TLR stimulation (Everts et al., 2014). We allowed for this in our study and only investigated the later metabolic reprogramming ~8 hours after activation. The data presented in this paper is the first to show that glucose represses DC functions at time points after activation when DC will be in draining lymph nodes. All other studies to date show glucose and glycolysis to be proinflammatory for immune cell function. Our data helps to clearly

delineate two distinct metabolic phases of Dc activation: early metabolic changes that will occur within the tissue (Ed Pearce's work) and later metabolic changes that will occur within lymph nodes. Our data argues that these later metabolic changes represent a metabolic regulatory axis that can be influenced by T cells in the lymph node to impact upon immune outcomes. We feel our paper, as a first of principle study, could open up exciting new areas of research in DC:T cell interaction within inflammatory lymph nodes.

Recently in the area of immunometabolism it has become appreciated that competition for nutrients can influence immune cell function, e.g. at tumor sites competition for glucose between tumors and T cells leads to reduced T cell activation (Chang et al., 2015). However up to this point studies in this area have been carried out during settings of immunological challenge, such as in tumour models. There has been no suggestion that competition for nutrients may be a mechanism for controlling immune responses within lymphoid tissue.

We hope you agree that our work presents a sufficient conceptual advance to justify publication.

Minor

1) Page 5; some of the Figure number in this section is incorrect. Please verify and edit as appropriate.

Reviewer #3 (Remarks to the Author):

There is an increasing appreciation for the role of metabolic programming in regulating the activation and differentiation of immune cells. This work serves to advance our knowledge by examining the role of glucose in repressing Dendritic Cell activation and the subsequent T cell responses. Specifically, the group demonstrates the ability of glucose availability to regulate glycolysis through HIF-1a and iNOS. Overall the data are robust and make exciting new connections. It would be great however, if the authors might address the following issues.

Figure 7 was very rewarding in that it took the findings regarding HIF1a, mTOR and iNOS and related them to the initial observations of the paper. However, I think it is crucial to do the same for low glucose levels (not just galactose). Meaning Figure 3d implies that lowering glucose levels achieves the same effect as culturing in galactose. Figure 3h tells us that maybe they aren't exactly equivalent. However, I think it is important that observations along the lines of Figure 7c,d,e, be shown for the low glucose levels

We agreed that we needed to show the effect of low glucose conditions and have added IL12a expression data at 24 hours for a range of glucose conditions that show similar results to galactose cultured GMDCs. Culturing long term in low glucose is not feasible as it is not really possible to regulate glucose levels for prolonged periods and prevent complete glucose deprivation.

Second, many aspects of Figure 8 cannot really be interpreted as robustly as the authors imply. Co-culturing T cells with DC's will have more effects than just depriving the DC's of glucose. Figure 8b is nice. Could the authors then take those populations and maybe simply show some differential molecular data for the 1:2 and 1:20 DC's?

We agree that there are additional effects of T cell:DC co-culture other than on glucose levels. In fact we have expanded our discussion to include arguments that other nutrients such as glutamine and leucine may also be distributed unequally between T cells and DC leading to the contrasting mTORC1 signalling in T cells and DC.

We are using this model to demonstrate, to our knowledge for the first time, nutrient competition between two immune cells during a physiological response and linking it to an immunological outcome. There remains much work to be done to fully understand the mechanisms involved but we feel that our work represents a significantly novel finding that could lead to greater understanding of DC induced T cell responses.

Performing signalling analysis on the 1:2 DC versus the 1:20 DC is problematic as once the DC:T cell clusters are disturbed to allow for purification of the DC, the DC are now exposed to nutrients and the signalling in the DC will be altered. Certainly, mTORC1 is rapidly activated by nutrients. The only real option is the use of confocal imaging approaches. However, this approach requires good specific antibodies and of the 4 commercial antibodies to HIF1 α that we tested, none were found to be suitable. All of the antibodies gave a strong signal by confocal imaging in HIF1a KO GM-DC.

We have included some pS6 imaging data that we believe provides extra proof of our proposed model. The pS6 imaging clearly shows the relationship between T cells clustering on DCs and the levels of pS6 within that DC. We believe this data strengthens the arguments made in Figure 8.

Andresen, L., Skovbakke, S.L., Persson, G., Hagemann-Jensen, M., Hansen, K.A., Jensen, H., and Skov, S. (2012). 2-deoxy D-glucose prevents cell surface expression of NKG2D ligands through inhibition of N-linked glycosylation. *J Immunol* *188*, 1847-1855.

Bouso, P., and Robey, E. (2003). Dynamics of CD8+ T cell priming by dendritic cells in intact lymph nodes. *Nat Immunol* *4*, 579-585.

Carroll, B., Maetzel, D., Maddocks, O.D., Otten, G., Ratcliff, M., Smith, G.R., Dunlop, E.A., Passos, J.F., Davies, O.R., Jaenisch, R., *et al.* (2016). Control of TSC2-Rheb signaling axis by arginine regulates mTORC1 activity. *eLife* *5*.

Chang, C.H., Qiu, J., O'Sullivan, D., Buck, M.D., Noguchi, T., Curtis, J.D., Chen, Q., Gindin, M., Gubin, M.M., van der Windt, G.J., *et al.* (2015). Metabolic Competition in the Tumor Microenvironment Is a Driver of Cancer Progression. *Cell* *162*, 1229-1241.

Everts, B., Amiel, E., Huang, S.C., Smith, A.M., Chang, C.H., Lam, W.Y., Redmann, V., Freitas, T.C., Blagih, J., van der Windt, G.J., *et al.* (2014). TLR-driven early glycolytic reprogramming via the kinases TBK1-IKK ϵ supports the anabolic demands of dendritic cell activation. *Nat Immunol* *15*, 323-332.

Everts, B., Amiel, E., van der Windt, G.J., Freitas, T.C., Chott, R., Yarasheski, K.E., Pearce, E.L., and Pearce, E.J. (2012). Commitment to glycolysis sustains survival of NO-producing inflammatory dendritic cells. *Blood* *120*, 1422-1431.

Krawczyk, C.M., Holowka, T., Sun, J., Blagih, J., Amiel, E., DeBerardinis, R.J., Cross, J.R., Jung, E., Thompson, C.B., Jones, R.G., *et al.* (2010). Toll-like receptor-induced changes in glycolytic metabolism regulate dendritic cell activation. *Blood* *115*, 4742-4749.

Pantel, A., Teixeira, A., Haddad, E., Wood, E.G., Steinman, R.M., and Longhi, M.P. (2014). Direct type I IFN but not MDA5/TLR3 activation of dendritic cells is required for maturation and metabolic shift to glycolysis after poly IC stimulation. *PLoS Biol* *12*, e1001759.

Rolf, J., Zarrouk, M., Finlay, D.K., Foretz, M., Viollet, B., and Cantrell, D.A. (2013). AMPK α 1: a glucose sensor that controls CD8 T-cell memory. *European journal of immunology* *43*, 889-896.

Sato, K., Ozaki, K., Oh, I., Meguro, A., Hatanaka, K., Nagai, T., Muroi, K., and Ozawa, K. (2007). Nitric oxide plays a critical role in suppression of T-cell proliferation by mesenchymal stem cells. *Blood* *109*, 228-234.

Starbeck-Miller, G.R., Xue, H.H., and Harty, J.T. (2014). IL-12 and type I interferon prolong the division of activated CD8 T cells by maintaining high-affinity IL-2 signaling in vivo. *J Exp Med* *211*, 105-120.

Tannahill, G.M., Curtis, A.M., Adamik, J., Palsson-McDermott, E.M., McGettrick, A.F., Goel, G., Frezza, C., Bernard, N.J., Kelly, B., Foley, N.H., *et al.* (2013). Succinate is an inflammatory signal that induces IL-1 β through HIF-1 α . *Nature* *496*, 238-242.

Valenzuela, J., Schmidt, C., and Mescher, M. (2002). The roles of IL-12 in providing a third signal for clonal expansion of naive CD8 T cells. *J Immunol* *169*, 6842-6849.

Yu, S.M., and Kim, S.J. (2010). Endoplasmic reticulum stress (ER-stress) by 2-deoxy-D-glucose (2DG) reduces cyclooxygenase-2 (COX-2) expression and N-glycosylation and induces a loss of COX-2 activity via a Src kinase-dependent pathway in rabbit articular chondrocytes. *Experimental & molecular medicine* *42*, 777-786.

Reviewers' comments:

Reviewer #1 (Remarks to the Author):

The authors have added new data and have amended the manuscript to sufficiently address most of my comments. However, my main comment regarding the relevance of their findings to DCs that do not express iNOS in response to TLR activation remains insufficiently addressed. I appreciate the fact that the authors have now added data to show that exogenous sources of NO can impact LPS induced costimulatory molecule expression, cytokine production and T cell priming by iNOS KO GMDCs in similar manner as endogenously derived NO does. While, this is clearly of added value, this was not the point I was hoping to be answered. The authors convincingly show that during DC-T cell interactions in vivo, competition for glucose, results in reduced mTOR activation in DCs, which they then link to impaired HIF1a and iNOS expression/NO production. This reduction in iNOS expression/NO production, then endows the DCs with a stronger T cell priming potential. However, cDCs express little to no iNOS and I have my doubts whether during T cell priming in a lymph node there will be sufficient NO produced by other cells to affect T cell priming in situ by cDCs. Or do they authors have direct evidence for this, that there can be sufficient NO production by other cells in lymph nodes to affect T cell polarization by DCs (such as inflammatory macrophages as the authors currently put forward as a potential source)?

So my main question is: does this relation between lower glucose consumption and improved T cell priming by DCs still hold up, when iNOS/NO is taken out of the equation? To address this the authors should repeat the experiment in which they cultured DCs and T cells at different ratios as shown in figure 8h, but now with iNOS KO DCs. If this experiment would show that also in the absence of NO there is lower T cell priming by DCs when you increase the ratio between T cells and DCs, then this would imply that the link between glucose uptake by DCs and their ability to prime T cell responses is applicable to T cell priming by DCs in general. This would improve the impact of their current findings. However, when there would be no difference anymore in T cell priming, then that would suggest that this axis is only operating in the context of iNOS expression. If the latter is the case, then it should be clearly stated in the abstract and discussion that this effect is seen only when T cell priming is driven by iNOS producing inflammatory DCs and/or is occurring strongly inflammatory LNs (NO derived from macrophages?). In that case the title should reflect that (for instance 'glucose represses T cell priming by inflammatory/iNOS expressing DCs')

Reviewer #2 (Remarks to the Author):

The authors have made some improvements to the manuscript. The addition of the NOS and HIF-1 k/o work is complimentary and supports their pharmacological data. I have two comments in regard to my previous critiques:

1) Regarding my original point #2 & main concern. I think it was important to examine arginase and appreciate that there could be multiple metabolites at play beyond glucose. However, the authors claim that glucose drives these dysfunctional DC phenotypes yet the authors now acknowledge in the discussion that glutamine, leucine and other amino acids could be equally important. The dilemma here is that HIF-1, NOS and mTOR regulates so many parallel and redundant metabolic pathways that the authors cannot rule out glucose as the main driver.

The authors also point that their "study is the first to describe the relationship between iNOS and HIF-1 in an immune cell subset under conditions of normoxia". I agree to a certain extent since the knockouts share some immunological/cytokine profiles (Fig. 7). The issue is that all of these data are somewhat descriptive and there is no clear mechanism how one gene affects the other in their DC model system. Because of the immunological similarities they observe, it suggest that there is considerable cross-talk between these factors on the metabolism of DCs yet the

manuscript does not address the problem of how this happens, which I feel is a key point. For instance, does iNOS drive the HIF-1 phenotype or vice versa? How does mTOR factor into this? Further, the authors point out in their response that "as a first of principle study, could open up exciting new areas of research in DC:T cell interaction within inflammatory lymph nodes". I certainly don't disagree this is an important area of investigation though they have limited data supporting their observations in DC's in vivo (see point below).

2) Regarding my original point #3. While I certainly acknowledge some of the technical limitations, this limits the impact of the work. For example, not being able to show that metabolic changes in HIF-1 or NOS k/o cells is a significant shortcoming towards in vivo relevance.

Overall, the paper is sound and covers an interesting topic. I would have liked to have seen more mechanistic insight of how all these regulators (HIF, mTOR, NOS) control the downstream metabolic features observed in DCs. I still have reservations regarding how the manuscript is a major conceptual advance given what is already published in the literature.

Reviewer #3 (Remarks to the Author):

The authors have addressed my questions.

Response to Reviewers' comments:

Reviewer 1

The authors have added new data and have amended the manuscript to sufficiently address most of my comments. However, my main comment regarding the relevance of their findings to DCs that do not express iNOS in response to TLR activation remains insufficiently addressed. I appreciate the fact that the authors have now added data to show that exogenous sources of NO can impact LPS induced costimulatory molecule expression, cytokine production and T cell priming by iNOS KO GMDCs in similar manner as endogenously derived NO does. While, this is clearly of added value, this was not the point I was hoping to be answered. The authors convincingly show that during DC-T cell interactions in vivo, competition for glucose, results in reduced mTOR activation in DCs, which they then link to impaired HIF1a and iNOS expression/NO production. This reduction in iNOS expression/NO production, then endows the DCs with a stronger T cell priming potential. However, cDCs express little to no iNOS and I have my doubts whether during T cell priming in a lymph node there will be sufficient NO produced by other cells to affect T cell priming in situ by cDCs. Or do they authors have direct evidence for this, that there can be sufficient NO production by other cells in lymph nodes to affect T cell polarization by DCs (such as inflammatory macrophages as the authors currently put forward as a potential source)?

So my main question is: does this relation between lower glucose consumption and improved T cell priming by DCs still hold up, when iNOS/NO is taken out of the equation? To address this the authors should repeat the experiment in which they cultured DCs and T cells at different ratios as shown in figure 8h, but now with iNOS KO DCs. If this experiment would show that also in the absence of NO there is lower T cell priming by DCs when you increase the ratio between T cells and DCs, then this would imply that the link between glucose uptake by DCs and their ability to prime T cell responses is applicable to T cell priming by DCs in general. This would improve the impact of their current findings. However, when there would be no difference anymore in T cell priming, then that would suggest that this axis is only operating in the context of iNOS expression. If the latter is the case, then it should be clearly stated in the abstract and discussion that this effect is seen only when T cell priming is driven by iNOS producing inflammatory DCs and/or is occurring strongly inflammatory LNs (NO derived from macrophages?). In that case the title should reflect that (for instance ' glucose represses T cell priming by inflammatory/iNOS expressing DCs')

(A) In response to the question of whether NO levels from inflammatory macrophages are likely to reach the levels required to affect cDC metabolism and function.

Firstly, it is certainly clear that NO is the key driver of the inhibition of OxPhos in GM-DC and inflammatory M1 macrophages and this is due to multiple actions of NO: (1) NO can compete for binding to cytochrome C oxidase (complex IV) directly inhibiting oxidative phosphorylation as suggested by Pearce et al's work (Everts et al., 2012), (2) NO has been linked to the break in the Krebs cycle that occurs between Citrate and alpha-ketoglutarate (unpublished data). Certainly NO has been shown to inhibit both aconitase and isocitrate dehydrogenase in other cell types (Gupta et al., 2012; Tortora et al., 2007; Yang et al., 2002). It is now well accepted that NO is crucial for inhibition of OxPhos in activated DC and Macrophages. Therefore, the fact that Pantel et al, demonstrated the characteristic inhibition of OxPhos in cDC, that did not express iNOS, stimulated *in vivo* following poly(I:C) injection strongly suggests that these cells were exposed to NO at sufficient levels to cause OxPhos inhibition (Pantel et al., 2014).

Multiple groups including the data we have added to this manuscript have shown that NO can affect cells in trans, i.e. NO produced by another cell in the local microenvironment (Amiel et al., 2014; Olekhovitch et al., 2014). Indeed, macrophages have been shown to be present in the T cell zone of lymph nodes (Asano et al., 2011).

(B) In response to the question of whether the mechanism that we describe will be relevant in cDC that lack iNOS if there is no exogenous source of NO. This is indeed an important question. We have performed the experiment that was suggested by reviewer 1 and now include this data in the manuscript(Fig.8i,j). The data show that in the absence of NO (iNOS^{KO} GM-DC) there is indeed increased T cell priming by DCs that are interacting with multiple T cells. This implies that the link between glucose availability to DCs and their ability to prime T cell responses is applicable to T cell priming by DCs in general. This experiment was an excellent suggestion and we believe that this data increases the impact of this study.

Reviewer 2

The authors have made some improvements to the manuscript. The addition of the NOS and HIF-1 k/o work is complimentary and supports their pharmacological data. I have two comments in regard to my previous critiques:

1) Regarding my original point #2 & main concern. I think it was important to examine arginase and appreciate that there could be multiple metabolites at play beyond glucose. However, the authors claim that glucose drives these dysfunctional DC phenotypes yet the authors now acknowledge in the discussion that glutamine, leucine and other amino acids could be equally important. The dilemma here is that HIF-1, NOS and mTOR regulates so many parallel and redundant metabolic pathways that the authors cannot rule out glucose as the main driver.

Wrt reviewer 2 comments about arginase in both this report and the original report. My interpretation is that reviewer 2 is suggesting that arginase is involved in this system somehow and is consuming arginine and differential levels of arginine are affecting T cell responses. The suggestion is that somehow with high T cell:DC ratios that arginase is becoming 'depleted' leading to decreased arginine metabolism by arginase and so increased arginine available for T cells. Therefore the T cell response is enhanced. Certainly, arginine and arginase have been linked to T cell responses (Dunand-Sauthier et al., 2014; Geiger et al., 2016).

There are a number of reasons why this mechanism is highly unlikely:

(a) GM-DC have low levels of Arg2 because it is negatively regulated by miR-155 (Dunand-Sauthier et al., 2014). GM-DCs have no arginine dependent effect on CD4 T cell proliferation unless miR-155 is deleted (Dunand-Sauthier et al., 2014). Therefore, in our cocultures with low T cell/DC ratios GM-DCs will not be able to inhibit T cell responses via arginine depletion and so the increased T cell responses at higher T cell/DC ratios cannot be due to release from such an inhibition.

(b) In Greiger et al's Cell paper they did detailed metabolic analyses and show that arginine levels become important between 24 and 48 hours post TCR stimulation. In our co-culture experiments (Fig 8h) we observed differences in T cell IFN γ production at 18 hours after addition to DCs (Geiger et al., 2016).

(c) As T cells are on the outside of the DC:T cell conjugates, they have access to the total media volume. Therefore arginine would need to be depleted from the

total media in order to affect T cells. This is contrast to the model we propose where the local DC microenvironment, and not the total media, becomes nutrient (glucose) deprived. For the coculture experiments the DC are plated at a low density (1×10^5 /ml) making it even more unlikely that they can deplete the media of all available arginine.

(d) The T cells engage in blastogenesis normally in the co-cultures with high T to DC ratios and the T cells have normal levels of mTORC1 activity (pS6 levels). It is known in CD8 T cells that mTORC1 signalling is partially sensitive to arginine depletion (personal correspondence with Doreen Cantrell's lab). Therefore, normal mTORC1 activity is suggestive of arginine sufficiency.

(e) Given that T cells also express Arg2 (Geiger et al., 2016) , it is highly unlikely that when 5 times more T cells (10:1 vs 2:1 ratios) are added to the cultures the total arginase activity is less.

Wrt reviewer 2 comments about glucose versus other nutrients.

We have described a signalling circuit that is sensitive to the levels of multiple nutrients, glucose and amino acids. In this study we have focused on glucose. We know that competition for glucose occurs between T cells and DC resulting in less glucose being available to DC in 10:1 T Cell:DC cultures. However, there are no tools yet available to prove competition for amino acids and glutamine and leucine between T cells and DC. The key point that our ms is making is that glucose represses DC function and glucose can become limiting for DC in DC-T cell conjugates. Whether the same is true for glutamine and leucine is an area of exciting continuing research.

I do not understand what reviewer 2 is saying in this statement "The dilemma here is that HIF-1, NOS and mTOR regulates so many parallel and redundant metabolic pathways that the authors cannot rule out glucose as the main driver?"

The authors also point that their "study is the first to describe the relationship between iNOS and HIF-1 in an immune cell subset under conditions of normoxia". I agree to a certain extent since the knockouts share some immunological/cytokine profiles (Fig. 7). The issue is that all of these data are somewhat descriptive and there is no clear mechanism how one gene affects the other in their DC model system. Because of the immunological similarities they observe, it suggest that there is considerable cross-talk between these factors on the metabolism of DCs yet the manuscript does not address the problem of

how this happens, which I feel is a key point. For instance, does iNOS drive the HIF-1 phenotype or vice versa? How does mTOR factor into this? Further, the authors point out in their response that “as a first of principle study, could open up exciting new areas of research in DC:T cell interaction within inflammatory lymph nodes”. I certainly don’t disagree this is an important area of investigation though they have limited data supporting their observations in DC’s in vivo (see point below).

We have done a detailed biochemical analysis of HIF1a and iNOS in DC. The data argues for significant cross-talk between these molecules not simply because there are immunological similarities in the respective KO mice. Figure 5 and 6 contain robust biochemical analysis of HIF1a and iNOS expression and activity following a whole range of perturbations, both pharmacological (SIETU, rapamycin, DMOG, NO donor), genetic (HIF1a Ko, and iNOS KO) and enzymatic (removal of the iNOS substrate arginine). In all these biochemical experiments perturbation of HIF1a activity inhibited iNOS expression and activity and perturbation of iNOS activity inhibited HIF1a expression. The data strongly argues that HIF1a and iNOS are interdependent.

With respect to which comes first iNOS or HIF1a, we have added some additional data to the manuscript that sheds some light on this question. We have done a time-course experiment looking at iNOS mRNA and PhD3 (as a measure of HIF1a activity) mRNA (see Supplementary Figure 2b,c). These data show that at early time-points iNOS mRNA expression is increased in an mTORC1 independent manner (rapamycin has no effect). At these early timepoints there is no PhD3 expression despite the fact that HIF1a mRNA consistently expressed from 4hrs-24hours post LPS stimulation (Suppl Fig.2a). However, at 16 hours post LPS there is a big increase in both iNOS and PhD3 expression and both these genes are now sensitive to rapamycin. These data argue that iNOS produced NO initially promotes HIF1a protein stabilization and then a feedforward loop ensues where HIF1a promotes iNOS mRNA expression and iNOS produced NO stabilized HIF1a protein. It is reported in the literature in cells other than DC that HIF1a can bind to a HRE (hypoxia response element) in the iNOS promoter to promote mRNA expression and that NO can induce HIF1a stabilization - we reference these articles in our ms:

“Indeed, in non-immune cells hypoxia-induced HIF1 α has been shown to bind to DNA elements in the iNOS promoter and increase gene expression, while NO has been reported to induce HIF1 α protein expression, though the mechanisms involved are not clear(Jung et al., 2000; Kasuno et al., 2004; Sandau et al., 2001).”

As to the question of how mTORc1 fits in. The time course data above argues that iNOS expression only becomes rapamycin sensitive once HIF1a is active, arguing that mTORC1 feeds into the HIF1a-iNOS signalling loop by controlling the expression of HIF1a protein. Though many of the best mTORC1 labs have investigated how exactly mTORC1 controls HIF1a using a variety of approaches including siRNA knockdown, expression mutated proteins, constitutively active proteins etc, there is still no real consensus of the exact mechanisms involved. Given that none of these techniques (siRNA, transfections etc) are possible in DC as they trigger DNA sensing pathways, this complex area is certainly beyond the scope of this project.

2) Regarding my original point #3. While I certainly acknowledge some of the technical limitations, this limits the impact of the work. For example, not being able to show that metabolic changes in HIF-1 or NOS k/o cells is a significant shortcoming towards in vivo relevance.

Measuring nutrient uptake and immune cell metabolism in vivo is what immunometabolism research groups aspires to do, but the technology to do this does not exist yet.

Overall, the paper is sound and covers an interesting topic. I would have liked to have seen more mechanistic insight of how all these regulators (HIF, mTOR, NOS) control the downstream metabolic features observed in DCs. I still have reservations regarding how the manuscript is a major conceptual advance given what is already published in the literature.

Reviewer 3

The authors have addressed my questions.

Amiel, E., Everts, B., Fritz, D., Beauchamp, S., Ge, B., Pearce, E.L., and Pearce, E.J. (2014). Mechanistic target of rapamycin inhibition extends cellular lifespan in dendritic cells by preserving mitochondrial function. *J Immunol* 193, 2821-2830.

Asano, K., Nabeyama, A., Miyake, Y., Qiu, C.H., Kurita, A., Tomura, M., Kanagawa, O., Fujii, S., and Tanaka, M. (2011). CD169-positive macrophages dominate antitumor immunity by crosspresenting dead cell-associated antigens. *Immunity* 34, 85-95.

Dunand-Sauthier, I., Irla, M., Carneseccchi, S., Seguin-Estevez, Q., Vejnar, C.E., Zdobnov, E.M., Santiago-Raber, M.L., and Reith, W. (2014). Repression of arginase-2 expression in dendritic cells by microRNA-155 is critical for promoting T cell proliferation. *Journal of immunology* (Baltimore, Md. : 1950) *193*, 1690-1700.

Everts, B., Amiel, E., van der Windt, G.J., Freitas, T.C., Chott, R., Yarasheski, K.E., Pearce, E.L., and Pearce, E.J. (2012). Commitment to glycolysis sustains survival of NO-producing inflammatory dendritic cells. *Blood* *120*, 1422-1431.

Geiger, R., Rieckmann, J.C., Wolf, T., Basso, C., Feng, Y., Fuhrer, T., Kogadeeva, M., Picotti, P., Meissner, F., Mann, M., et al. (2016). L-Arginine Modulates T Cell Metabolism and Enhances Survival and Anti-tumor Activity. *Cell* *167*, 829-842 e813.

Gupta, K.J., Shah, J.K., Brotman, Y., Jahnke, K., Willmitzer, L., Kaiser, W.M., Bauwe, H., and Igamberdiev, A.U. (2012). Inhibition of aconitase by nitric oxide leads to induction of the alternative oxidase and to a shift of metabolism towards biosynthesis of amino acids. *J Exp Bot* *63*, 1773-1784.

Jung, F., Palmer, L.A., Zhou, N., and Johns, R.A. (2000). Hypoxic regulation of inducible nitric oxide synthase via hypoxia inducible factor-1 in cardiac myocytes. *Circ Res* *86*, 319-325.

Kasuno, K., Takabuchi, S., Fukuda, K., Kizaka-Kondoh, S., Yodoi, J., Adachi, T., Semenza, G.L., and Hirota, K. (2004). Nitric oxide induces hypoxia-inducible factor 1 activation that is dependent on MAPK and phosphatidylinositol 3-kinase signaling. *J Biol Chem* *279*, 2550-2558.

Olekhovitch, R., Ryffel, B., Muller, A.J., and Bousso, P. (2014). Collective nitric oxide production provides tissue-wide immunity during *Leishmania* infection. *J Clin Invest* *124*, 1711-1722.

Pantel, A., Teixeira, A., Haddad, E., Wood, E.G., Steinman, R.M., and Longhi, M.P. (2014). Direct type I IFN but not MDA5/TLR3 activation of dendritic cells is required for maturation and metabolic shift to glycolysis after poly IC stimulation. *PLoS Biol* *12*, e1001759.

Sandau, K.B., Fandrey, J., and Brune, B. (2001). Accumulation of HIF-1alpha under the influence of nitric oxide. *Blood* *97*, 1009-1015.

Tortora, V., Quijano, C., Freeman, B., Radi, R., and Castro, L. (2007). Mitochondrial aconitase reaction with nitric oxide, S-nitrosoglutathione, and peroxynitrite: mechanisms and relative contributions to aconitase inactivation. *Free Radic Biol Med* *42*, 1075-1088.

Yang, E.S., Richter, C., Chun, J.S., Huh, T.L., Kang, S.S., and Park, J.W. (2002). Inactivation of NADP(+)-dependent isocitrate dehydrogenase by nitric oxide. *Free Radic Biol Med* *33*, 927-937.

REVIEWERS' COMMENTS:

Reviewer #1 (Remarks to the Author):

The addition of the T cell priming data with the iNOS KO GMDCs is definitely of added value (fig 8 i/j), but I think they are also open for an alternative explanation for why in the absence of iNOS/NO there is still an increased T cell priming at lower DC/T cell ratios. Couldn't it be that at high DC/T cell ratios there is a high consumption of glucose by long-lived iNOS KO DCs, that by the time you do the ICS for IFNG (after two days) there is sufficient deprivation of glucose to limit effective IFNG secretion (Chang et al, cell 2013), while at lower ratios, with fewer DCs in the culture, there is less glucose consumption by the DCs and hence more efficient IFNG secretion by the T cells.

Could the authors comment on this?

Other than that I have no further comments.

Response to Reviewers' comments:

Reviewer #1 (Remarks to the Author):

The addition of the T cell priming data with the iNOS KO GMDCs is definitely of added value (fig 8 i/j), but I think they are also open for an alternative explanation for why in the absence of iNOS/NO there is still an increased T cell priming at lower DC/T cell ratios. Couldn't it be that at high DC/T cell ratios there is a high consumption of glucose by long-lived iNOS KO DCs, that by the time you do the ICS for IFNG (after two days) there is sufficient deprivation of glucose to limit effective IFNG secretion (Chang et al, cell 2013), while at lower ratios, with fewer DCs in the culture, there is less glucose consumption by the DCs and hence more efficient IFNG secretion by the T cells.

Could the authors comment on this?

Other than that I have no further comments.

Response: The numbers of DC in the co-cultures are fixed and it is the numbers of T cells that are varied to alter the DC:T cell ratio. Therefore, the alternative explanation suggested by reviewer 1 is not a possibility.